# Beyond Hard Supervised Fine-tuning: Enhancing Image–text Alignment of Strong Models with Weak Models

## Abstract

Image–text alignment models, such as CLIP, are typically trained with large-scale contrastive learning: paired data are treated as positives, while all unpaired pairs are treated as negatives. However, this hard supervision overlooks the fact that some unpaired pairs are semantically related rather than irrelevant, and penalising them as strict negatives introduces noise that limits model performance. We propose Permute-then-Adapt (PTA), a weak-to-strong supervision framework that addresses this issue. PTA comprises two key innovations: (1) a permutation-based thresholding that identifies and filters unreliable negatives by estimating a null distribution of similarities, and (2) a soft supervision strategy that leverages above-threshold similarities to provide extra training signals. Across benchmarks, PTA consistently improves the alignment ability of strong models on object recognition and cross-modal retrieval.

## 1 Introduction

Recent advances in multimodal learning (Radford et al., 2021; Jia et al., 2021; Mu et al., 2022; Xu et al., 2023; Fang et al., 2023) have demonstrated the immense potential of image-text alignment models for understanding and connecting visual and textual information. Models like CLIP (Radford et al., 2021) have shown remarkable capabilities in zero-shot recognition (Deng et al., 2009), cross-modal retrieval (Lin et al., 2014), and serving as foundational components for larger multimodal systems (Wang et al., 2024). These models achieve their performance through contrastive learning on large-scale image-text pairs, where the core training objective is to maximise similarity between matched pairs while minimising similarity between unmatched ones.

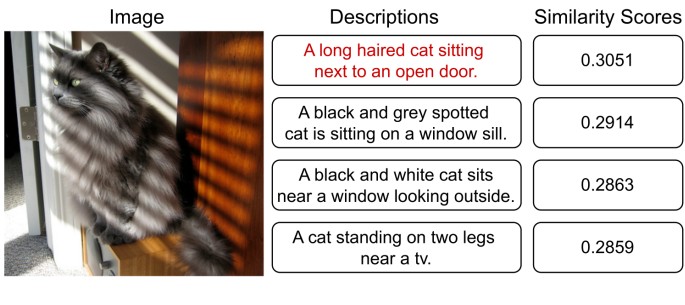

| Image | Descriptions | Similarity Scores |
|---|---|---|
| | A long haired cat sitting next to an open door. | 0.3051 |
| | A black and grey spotted cat is sitting on a window sill. | 0.2914 |
| | A black and white cat sits near a window looking outside. | 0.2863 |
| | A cat standing on two legs near a tv. | 0.2859 |

Figure 1: A sample showing an image of a long-haired cat by a door, where red description is the ground truth caption. Others are retrieved as similar captions. These alternatives demonstrate semantic similarity while capturing different visual aspects of the scene, but are regarded as negative paired texts.

However, a critical challenge in training these models lies in the fundamental assumption about unmatched pairs: current approaches (Radford et al., 2021) treat all non-paired images and text as completely unrelated, assigning them binary negative labels during training. This assumption oversimplifies the rich semantic relationships that may exist between different modalities. For instance, in Figure 1, we use CLIP-B/32 (Radford et al., 2021) on the CoCo dataset (Lin et al., 2014), retrieving the top-$k$ text candidates for an image of a "cat sitting next to an open door." Although these retrieved captions are semantically relevant (e.g., mentioning a cat or indoor scenes), the standard contrastive objective forces them to be treated as negatives, despite their obvious overlap with the visual content. This illustrates how the binary labelling scheme discards

useful semantic information. To further validate this issue, We empirically validate this phenomenon in section 3.1, where we show that the prevalence of false negatives grows significantly with batch size (Figure 2).

Furthermore, the quality of learned representations heavily depends on the reliability of these binary supervision signals. When training on web-scale data, the assumption that paired data is always semantically aligned while unpaired data is always unrelated becomes increasingly problematic. Research has shown that web-crawled image-text pairs often contain noise, misalignments, and varying degrees of semantic relevance (Kang et al., 2023; Gadre et al., 2024). The binary supervision paradigm fails to capture these nuances, potentially introducing noise into the training process of existing large-scale models and limiting the model's ability to have better representations.

Recent efforts have attempted to mitigate these issues by filtering web-scale image-text pairs using fixed thresholds on similarity scores produced by pre-trained CLIP models. For example, Data Filtering Networks (Fang et al., 2023) adopt a cosine similarity threshold of $0.3$ to decide whether an image-text pair should be retained. While such heuristics can improve dataset quality, they remain sensitive to the choice of threshold and may discard semantically relevant pairs or retain noisy ones. This limitation motivates the need for more adaptive and statistically grounded approaches that can better capture the nuanced distribution of semantic relevance in large-scale data. In Appendix A.4, we review more relevant work in detail.

In this paper, we propose a novel framework for enhancing the image-text alignment ability of a strong model by leveraging the guidance of a weak model. Our motivation stems from the observation that, although weak models (e.g., smaller CLIP variants) are limited in overall accuracy, they still encode useful signals about semantic relatedness between image-text pairs. Rather than relying on a single fixed threshold, we treat the weak model as a noisy but informative supervisor whose similarity scores can reveal which pairs are potentially semantically relevant. In addition, weak models are typically lightweight and computationally efficient, making them practical supervisors for guiding stronger models at scale. To robustly extract this information, we introduce Permute-then-Adapt (PTA), a *weak-to-strong supervision* framework, which employ the thresholding mechanism inspired by the permutation test in statistics, which automatically filters out unreliable signals by comparing them against the null distribution of randomly permuted pairs. This approach requires no additional validation data, adapts naturally across datasets, and transforms traditional contrastive learning by enriching binary supervision with soft, statistically grounded guidance. We also propose a soft supervision strategy that leverages above-threshold similarities to provide graded training signals. Together, these components enrich traditional binary supervision, allowing multiple semantically valid associations for each image or text and leading to more nuanced and robust representations. We validate our approach through extensive experiments across seven image classification and two retrieval benchmarks, demonstrating consistent improvements over strong baselines.

Our main contributions can be summarised as follows: (i) We introduce **Permutation-based Thresholded Adaptation (PTA)**, a novel refinement mechanism for weak supervision that adaptively determines the reliability of similarity signals without requiring extra validation data. (ii) We propose a **soft supervision strategy** that leverages above-threshold similarities to provide graded training signals, enriching traditional binary contrastive learning with multiple semantically valid associations. (iii) Through extensive experiments on seven image classification tasks and two retrieval benchmarks, we show that PTA consistently outperforms existing weakly supervised baselines and exhibits superior robustness to noisy supervision.

## 2 PRELIMINARIES

**Notations.** Throughout this paper, we use lowercase bold letters (e.g., $\mathbf{x}$, $\mathbf{v}$) to denote vectors, uppercase bold letters (e.g., $\mathbf{W}$) for matrices, calligraphic letters (e.g., $\mathcal{D}$) for sets and distributions, $|\cdot|$ for $l^2$ norm, and $\langle \cdot, \cdot \rangle$ for inner product; specifically, $\mathbb{R}^d$ denotes the $d$-dimensional Euclidean space, $\mathcal{X}$ represents the space of images, and $\mathcal{Z}$ represents the space of all possible text descriptions.

**Dual Encoder Models.** Dual encoder models (Radford et al., 2021) belong to a class of neural architectures designed to learn representations of two different modalities in a shared embedding space. These models consist of two separate encoders that process different input types (e.g., images and text) and project them into a common semantic space where similarity can be directly computed.

Formally, we define: an image encoder $f_v : \mathcal{X} \to \mathbb{R}^d$ that maps images from space $\mathcal{X}$ to a $d$-dimensional embedding, a text encoder $f_t : \mathcal{Z} \to \mathbb{R}^d$ that maps text from space $\mathcal{Z}$ to the same $d$-dimensional space, and a similarity function $s : \mathbb{R}^d \times \mathbb{R}^d \to [-1, 1]$ that measures the alignment between embeddings.

**Image-text Alignment.** Image-text alignment (Li et al., 2024) refers to the task of learning representations that capture semantic correspondences between visual and textual content. Given a dataset $\mathcal{D} = \{(I_i, T_i)\}_{i=1}^N$ of image-text pairs, where $I_i \in \mathcal{X}$ represents an image and $T_i \in \mathcal{Z}$ represents its corresponding text description, the goal is to learn a mapping that reflects semantic similarity in a shared embedding space.

The alignment task can be completed based on the dual encoder framework. Given a batch of $B$ image-text pairs $\mathcal{D}_{\text{batch}} = \{(I_k, T_k)\}_{k=1}^B \subset \mathcal{D}$, contrastive learning aligns embeddings by maximising similarity between matched pairs (e.g., $I_1$ and $T_1$) while minimising similarity for mismatched pairs (e.g., $I_1$ and $T_2$ that are not paired in the dataset $\mathcal{D}$). Let $\mathbf{v}_k = f_v(I_k) \in \mathbb{R}^d$ and $\mathbf{t}_m = f_t(T_m) \in \mathbb{R}^d$ denote the normalised embeddings (i.e., $|\mathbf{v}_k| = |\mathbf{t}_m| = 1$) of image $I_k$ and text $T_m$, respectively. The similarity between $\mathbf{v}_k$ and $\mathbf{t}_m$ is measured via inner product $\langle \mathbf{v}_k, \mathbf{t}_m \rangle$, scaled by temperature $Q > 0$. For each anchor $k$, the image-to-text and text-to-image contrastive losses are

$$\ell_{\text{i2t}}^{(k)} = - \log \frac{\exp\left(\langle \mathbf{v}_k, \mathbf{t}_k \rangle / Q\right)}{\sum_{m=1}^B \exp\left(\langle \mathbf{v}_k, \mathbf{t}_m \rangle / Q\right)}, \; \ell_{\text{t2i}}^{(k)} = - \log \frac{\exp\left(\langle \mathbf{t}_k, \mathbf{v}_k \rangle / Q\right)}{\sum_{m=1}^B \exp\left(\langle \mathbf{t}_k, \mathbf{v}_m \rangle / Q\right)}. \tag{1}$$

The total loss

$$\mathcal{L}_{\text{batch}} = \frac{1}{2B} \sum_{k=1}^B \left( \ell_{\text{i2t}}^{(k)} + \ell_{\text{t2i}}^{(k)} \right) \tag{2}$$

is minimised over $\mathcal{D}$ to align the joint embedding space, where in-batch negative pairs ($k \neq m$) enforce discriminative separation.

# 3  IMAGE-TEXT ALIGNMENT WITH WEAK MODELS

In this section, we will show why weak models could be beneficial for image-text alignment and propose our method to enhance the image-text alignment of a strong model with weak models. We will first revisit what the actual training dataset is when we train a dual encoder model for image-text alignment and motivate our method.

## 3.1  MOTIVATION: THE FALSE-NEGATIVE PROBLEM IN CONTRASTIVE LEARNING

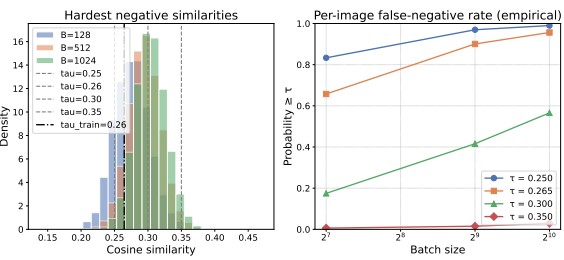

In Section 2, we briefly show how previous methods train a dual encoder model for image-text alignment, which involves paired data and non-paired data. Here, we will use a function $l_{\mathcal{D}}(I, T)$ regarding the paired image-text dataset $\mathcal{D}$ (shown below) to formalise the actual training dataset used for training the dual encoder model.

Figure 2: Empirical analysis of false negatives on Flickr30k. **Left:** Histogram of the hardest in-batch negative similarities across batch sizes, with thresholds $Q \in \{0.25, 0.2645, 0.30, 0.35\}$. **Right:** Per-image false-negative rate vs. batch size. The prevalence of false negatives grows rapidly with $B$, reaching over 90% at $Q_{\text{train}} = 0.2645$ and $B = 1024$.

$$l_{\mathcal{D}}(I, T) = \begin{cases} 1 & \text{if } (I, T) \in \mathcal{D}, \\ 0 & \text{otherwise.} \end{cases} \tag{3}$$

With the above function $l_{\mathcal{D}}$ and the paired image-text dataset $\mathcal{D}$, the training set for the image-text alignment is

$$\mathcal{D}_{\text{act}}(\mathcal{D}) = \{(I_i, T_j, l_{\mathcal{D}}(I_i, T_j))\}_{i,j=1}^N, \tag{4}$$

where $\ell_{\mathcal{D}}(I_i, T_j) \in \{0, 1\}$ denotes the binary label indicating whether image $I_i$ and text $T_j$ form a positive pair. In other words, $\mathcal{D}_{\text{act}}(D)$ represents the set of all image-text pairs in $D$ annotated with

their supervision signal. Training with $\mathcal{D}_{\text{act}}$ therefore reduces to a binary classification problem using the loss $\mathcal{L}_{\text{batch}}$ in Eq. 2. Thus, when we train the dual encoder model for image-text alignment, it is actually a binary classification problem with a loss $\mathcal{L}_{\text{batch}}$ in Eq. equation 2.

A key limitation of this formulation is that all unpaired pairs are treated as negatives. While this assumption holds in standard classification, it is problematic in image-text alignment: unpaired data may still share semantic relevance (e.g., overlapping objects or attributes). For instance, even if $I_1$ is paired with $T_1$, it may also be semantically close to $T_2$ (see Figure 3). Forcing $\ell_{\mathcal{D}}(I_1, T_2) = 0$ introduces a *false negative*, distorting the representation space. This paper addresses how to mitigate such incorrect supervision signals.

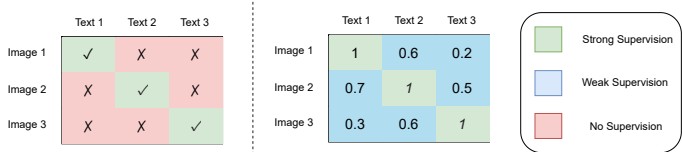

Figure 3: Comparison of supervision strategies in image-text alignment tasks. Left: traditional binary supervision in CLIP. Right: our proposed continuous weak supervision with partial alignment signals.

**Empirical evidence.** To quantify the prevalence of false negatives, we compute the cosine similarity between each image and all in-batch negatives using a pre-trained CLIP B/32 model. Figure 2 (left) plots the distribution of the hardest negative similarity for different batch sizes. As the batch size grows (from $B = 128$ to $B = 1024$), the distribution shifts to the right, indicating a higher chance of encountering semantically related negatives. The dashed vertical lines mark similarity thresholds $Q \in \{0.25, 0.2645, 0.30, 0.35\}$, with $Q_{\text{train}} = 0.2645$ (black dash-dot) denotes the threshold used in our training procedure.

**Prevalence increases with batch size.** Figure 2 (right) reports the per-image false-negative rate, defined as the probability that an image has at least one negative with similarity above $Q$. At $Q_{\text{train}} = 0.2645$, the false-negative rate increases from $\sim 0.67$ at $B = 128$ to over 0.90 at $B = 1024$. Even with a more conservative threshold ($Q = 0.30$), more than half the images encounter false negatives at $B = 512$.

**Implication.** These findings demonstrate that false negatives are not rare corner cases but systematically arise under realistic batch sizes. As illustrated in Figure 3, traditional binary supervision treats all in-batch negatives as equally uninformative, which can lead to plausible negatives being mislabelled. In contrast, our method introduces continuous weak supervision with partial alignment signals, explicitly identifying and mitigating the impact of such plausible negatives. This motivates the need for our proposed approach, which reduces the risk of false-negative bias in image-text alignment tasks.

### 3.2 Our Method: Permute-then-Adapt

In this section, we will discuss problem setup and our method. Figure 4 demonstrates our method visually.

**Problem Setup.** Since the issue demonstrated in Section 3.1 probably exists in the pre-training process of some large-scale models (a.k.a. strong models) (Radford et al., 2021), in this paper, we are particularly interested in the problem of *enhancing existing strong models by using a more accurate signal to further finetune them*. Inspired by weak-to-strong supervision (Burns et al., 2023), we mainly investigate the situation where the more accurate signal comes from a relatively small model (a.k.a. a weak model), which is low-cost during inference (Burns et al., 2023).

**Weak teacher similarities**. To have a more accurate signal from weak models, a straightforward way is to replace $l_{\mathcal{D}}$ with $l_f$:

$$l_f(I, T; Q) = \begin{cases} 1 & \text{if } f(I, T) > Q, \\ 0 & \text{otherwise,} \end{cases} \tag{5}$$

where $f$ is a weak model that is able to give a similarity score for non-paired data (e.g., the similarity score between $I_1$ and $T_2$) and $Q$ is a threshold to keep the most similar pairs. The weak model $f$ is also one of dual-encoder models, containing three parts: an image encoder $f_v$, a text encoder $f_t$, and a scoring function $s$. The output of $f(I, T)$ is equal to $s(f_v(I), f_t(T))$. With $l_f$, we can have a

new training dataset related to the choice of $Q$:

$$\mathcal{D}_{\text{act}}^{f}(Q) = \{(I_i, T_j, l_f(I_i, T_j; Q))\}_{i,j=1}^{N}. \tag{6}$$

Then, we can further fine-tune a strong model with $\mathcal{D}_{\text{act}}^{f}(Q)$[1]. It is clear that the $\mathcal{D}_{\text{act}}^{f}(Q)$ in Eq. 6 is heavily dependent on the selection $Q$. However, this is not ideal when we want to use it in practice, because the ideal value of $Q$ might change for different paired datasets and the meaning of $Q$'s values is also not consistent across different paired datasets. In the following, we will address this issue by leveraging the concept of significance level $\alpha$ in permutation testing that is frequently used in statistical hypothesis testing and works on popular test statistics (Hemerik & Goeman, 2018).

### 3.2.1 PERMUTE: ESTIMATE THE NULL DISTRIBUTION OF SIMILARITY SCORES OF IMAGE-TEXT PAIRS

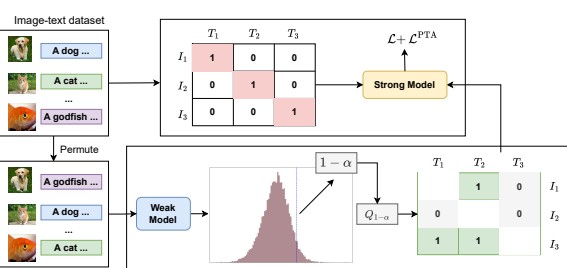

Figure 4: Overview of our Permute-then-Adapt framework for enhancing image-text alignment. Starting from an original dataset (left), we obtain weak supervision signals from a weak model after shuffling. The weak model's outputs are analysed through a permutation-based threshold $Q_{1-\alpha}$ derived from the null distribution. The resulting refined similarity matrix is then used to guide the training of a strong model through our refined weak supervision loss $\mathcal{L}_{\text{weak}}$.

**Permutation-based Null Distribution**. From the above analysis, it can be seen that we need to select a proper $Q$ such that we can distinguish between relevant and irrelevant image-text pairs. Thus, it is necessary to know what the similarity score should be for irrelevant image-text pairs. To estimate this score, we first recall a fact for two independent random variables (Gretton et al., 2007): if two random variables $X$ and $Y$ are independent, then, for their observations $\{(x_i, y_i)\}_{i=1}^{n}$, even after we shuffle one variable's observations, the shuffled observations $\{(x_i, y_i')\}_{i=1}^{n}$ and the original observations are still from the same distribution, where $y_i'$ is the observation after the shuffle. Based on this fact, permutation testing is used to estimate the null distribution of a test statistic on given observations (Xu et al., 2024), where the null distribution refers to a scenario in which we assume two random variables are independent. If the value of the test statistic on original observations falls within high-density areas of the null distribution (e.g., this value is close to the median value of the null distribution), we can say two random variables are independent.

The idea of permutation testing is particularly relevant when assessing the relationship between images and their corresponding textual descriptions. By shuffling the text components and computing similarity scores (via a weak model) for these permuted pairs, one can establish a baseline (i.e., a null distribution) that represents the scenario where images and texts are unrelated. Comparing the similarity score of any image-text pair against this baseline allows for the *determination of whether the observed association (i.e., they are paired) is statistically significant or likely due to chance*.

Therefore, we construct an empirical null distribution of the similarity scores by shuffling the text descriptions in the paired dataset $\mathcal{D}$. Let $\mathcal{G}$ be a group of transformations on $\mathcal{T} = (T_1, \ldots, T_N)$. In this paper $\mathcal{G} = \mathfrak{G}_N$, denoting the permutation group of $[N]$ by $\mathfrak{G}_N$ and its elements by $\sigma \in \mathfrak{G}_N$, $\sigma : [N] \rightarrow [N]$, where $[N] = \{1, \ldots, N\}$. Then, we can write the permutation on $\mathcal{T}$ as $g\mathcal{T}$, an action of $g \in \mathcal{G}$ on $\mathcal{T}$; e.g. defining $\sigma\mathcal{T} = (T_{\sigma(1)}, \ldots, T_{\sigma(N)})$ for the group elements $\sigma \in \mathfrak{G}_N$. Hence, the null distribution of the similarity scores on $\mathcal{D}$ is[2]

$$\mathcal{P}_{\text{null}} = \{f(I_i, T_{\sigma(i)}) : (I_i, T_i) \in \mathcal{D}\}. \tag{7}$$

A fixed global threshold fails to transfer across datasets or weak models, whereas a permutation-based cutoff adapts automatically to the scale and calibration of the weak similarities.

---

[1]Although Eq. 6 defines the full candidate set of $N^2$ image–text pairs, in practice we do not materialise all possible pairs during training. Instead, training operates on mini-batches $\mathcal{B} \subseteq \mathcal{D}$, where each image $I_k \in \mathcal{B}$ contributes its true caption $T_k$ (positive) and a subset of randomly sampled negatives $T_m$ with $m \neq k$.

[2]As noted in Appendix A.3, this calibration step is done once before training and demonstrates a low cost.

### 3.2.2 ADAPT: SELECT $(1-\alpha)$-TH PERCENTILE VALUE AS THE THRESHOLD $Q$

**Statistical Threshold with Type-I Error Control.** Recall that $\mathcal{P}_{\text{null}}$ denotes the empirical null distribution of weak similarities. Intuitively, $\mathcal{P}_{\text{null}}$ captures how large the weak similarity $f(I, T)$ can be when $I$ and $T$ are *unrelated*. Given this distribution, we select a threshold by specifying a user-controlled significance level $\alpha \in (0, 1)$.

Concretely, we choose the $(1 - \alpha)$-th percentile of $\mathcal{P}_{\text{null}}$ as the threshold $Q$. This plays the role of a type-I error control: under the null hypothesis that $I$ and $T$ are independent, the probability that a random mismatched pair exceeds $Q$ is at most $\alpha$. Formally, the $\eta$-th percentile of $\mathcal{P}_{\text{null}}$ is defined as

$$\text{percentile}_{\eta}(\mathcal{P}_{\text{null}}) = \min\left\{ r \in \mathbb{R} \ \middle| \ \frac{1}{N} \sum_{i=1}^{N} \mathbf{1}(s_i \leq r) \geq \eta, s_i \in \mathcal{P}_{\text{null}}^{\text{sorted}} \right\}, \tag{8}$$

where $\mathcal{P}_{\text{null}}^{\text{sorted}}$ is the sorted set of samples from $\mathcal{P}_{\text{null}}$ (in increasing order), and $\mathbb{1}(\cdot)$ is the indicator function. Given a significance level $\alpha$, we set

$$Q = \text{percentile}_{1-\alpha}(\mathcal{P}_{\text{null}}), \tag{9}$$

so that at most an $\alpha$ fraction of truly unrelated pairs are (falsely) treated as *suspicious* under the null. We then use this threshold to generate labels for an unpaired image–text pair $(I, T)$:

$$l_f^{\text{p}}(I, T; \alpha) = \begin{cases} 1 & \text{if } f(I, T) > \text{percentile}_{1-\alpha}(\mathcal{P}_{\text{null}}), \\ 0 & \text{otherwise}, \end{cases} \tag{10}$$

and collect these pseudo-labels into a training set of additional (soft) positives for fine-tuning the strong model,

$$\mathcal{D}_{\text{PTA}}^{f}(\alpha) = \{(I_i, T_j, l_f^{\text{p}}(I_i, T_j; \alpha))\}_{i,j=1; i \neq j}^{N}. \tag{11}$$

In other words, $\alpha$ directly specifies the tolerated type-I error rate on the permuted null similarities, and $Q$ is a statistically calibrated cutoff that converts weak scores into extra multi-positive supervision for the strong model. Considering a subset $\mathcal{K} \subset [N]$ whose size is $B$, for each mini-batch the training batch is a union between paired data from $\mathcal{D}$ (called $\mathcal{D}_{\text{batch}}$) and unpaired data with PTA labels $l_f^{\text{p}}$ (called $\mathcal{D}_{\text{batch}}^{\text{PTA}}$). Specifically,

$$\mathcal{D}_{\text{batch}} = \{(I_k, T_k) : k \in \mathcal{K}\}, \ \mathcal{D}_{\text{batch}}^{\text{PTA}} = \{(I_k, T_m, l_f^{\text{p}}(I_k, T_m; \alpha)) : k, m \in \mathcal{K}\}. \tag{12}$$

It is clear that $\mathcal{D}_{\text{batch}}^{\text{PTA}} \subset \mathcal{D}_{\text{PTA}}^{f}(\alpha)$ (for simplicity, we omit $\alpha$ and $f$ in the notation regarding the mini-batch). In the following, we use a shortcut $l_{km}^{\text{PTA}}$ to represent $l_f^{\text{p}}(I_k, T_m; \alpha)$ which is the PTA-generated label for pair $(I_k, T_m)$.

**Multi-positive InfoNCE.** Given a strong model $f^s$, the PTA loss turns each anchor into a *multi-positive* objective (Xu et al., 2017): for an image anchor $I_k$, all texts $T_m$ with $\ell_{km}^{\text{PTA}} = 1$ are treated as additional positives in the batch, and the remaining texts are negatives. The image-to-text loss for the $k$-th image is

$$\ell_{\text{i2t}}^{\text{PTA},(k)} = -\log \sum_{m=1}^{B} l_{km}^{\text{PTA}} \frac{\exp(f^s(I_k, T_m))}{\sum_{m=1}^{B} \exp(f^s(I_k, T_m))}, \tag{13}$$

where $f^s(I_k, T_m)$ is the similarity (logit) produced by $f^s$. When each row has exactly one positive with $\ell_{km}^{\text{PTA}} = 1$, Eq. 13 reduces to the standard CLIP image-to-text InfoNCE loss. PTA generalizes this to the dynamic multi-positive case where the positive set $\{m : \ell_{km}^{\text{PTA}} = 1\}$ is inferred from the weak teacher via the permutation-calibrated threshold. Similarly, for a text anchor $T_m$, all images $I_k$ with $\ell_{km}^{\text{PTA}} = 1$ form the positive set, and the text-to-image loss is

$$\ell_{\text{t2i}}^{\text{PTA},(m)} = -\log \sum_{k=1}^{B} l_{km}^{\text{PTA}} \frac{\exp(f^s(I_k, T_m))}{\sum_{k=1}^{B} \exp(f^s(I_k, T_m))}. \tag{14}$$

**PTA loss.** Thus, the PTA loss on the batch $\mathcal{D}_{\text{batch}}^{\text{PTA}}$ is

$$\mathcal{L}_{\text{batch}}^{\text{PTA}} = \frac{1}{B} \sum_{k=1}^{B} \ell_{\text{i2t}}^{\text{PTA},(k)} + \frac{1}{B} \sum_{m=1}^{B} \ell_{\text{t2i}}^{\text{PTA},(m)}. \tag{15}$$

This can be viewed as a supervised multi-positive contrastive loss, where the additional positives are not manually annotated but are selected automatically from in-batch negatives using the permutation-calibrated weak similarities. The final integrated loss combines the contrastive loss $\mathcal{L}_{\text{batch}}$ and the refined weak loss $\mathcal{L}_{\text{batch}}^{\text{PTA}}$:

$$\mathcal{L}_{\text{batch}}^{\text{total}} = \underbrace{\mathcal{L}_{\text{batch}}}_{\text{Standard contrastive}} + \lambda \underbrace{\mathcal{L}_{\text{batch}}^{\text{PTA}}}_{\text{Supervision from a weak model}}, \tag{16}$$

where standard contrastive term ($\mathcal{L}_{\text{batch}}$) ensures strict alignment of ground-truth pairs ($k = m$) and penalizes all in-batch negatives ($k \neq m$). The PTA term ($\mathcal{L}_{\text{batch}}^{\text{PTA}}$) guides the model to additionally align pairs $(I_k, T_m)$ whose $l_{km}^{\text{PTA}} = 1$, which are statistically significant but non-ground-truth. This introduces soft positives into the training process, enriching the semantic structure of the embedding space. $\lambda$ represents the weight between 0 and 1. This integrated approach enables the model to *simultaneously* learn from hard negatives (via $\mathcal{L}_{\text{batch}}$) and statistically meaningful soft positives (via $\mathcal{L}_{\text{batch}}^{\text{PTA}}$), bridging the gap between binary supervision and signals from a weak model. The result is an embedding space that captures both semantic relationships and robust alignment of ground-truth pairs. We detailed how we pick the threshold in Appendix A.3.

**Partial Fine-tuning of CLIP.** In addition to our proposed PTA objective, we fine-tune only the last few transformer layers of CLIP while freezing earlier ones. This design is based on the observation that lower layers primarily capture general modality-specific features, whereas higher layers concentrate on cross-modal alignment signals. Updating only the top layers therefore balances stability and adaptivity: the model preserves transferable low-level representations while adapting high-level semantics for improved alignment. Recent work on multi-modal adaptation, such as the Multi-Modal Adapter framework (Yang et al., 2024), similarly emphasises that cross-modality interaction is most effective when applied in later layers. Our approach follows this principle by restricting updates to the top $L$ layers, which leads to improved alignment without the risk of overfitting associated with full fine-tuning. The implementation details can be found in Appendix A.3.

## 4 EXPERIMENTS

**Datasets and Baselines.** We established a new benchmark in the current setting. Datasets and baselines can be found in Appendix A.1 and Appendix A.2. The implementation details can be found in Appendix A.3.

**Main results.** Table 1 compares zero-shot accuracies after fine-tuning on COCO. Fine-tuning the strong student with our PTA consistently outperforms baselines that either fine-tune CLIP directly (CLIP-B16+SFT) or incorporate alternative teacher signals. Using a ViT-B/32 weak teacher and a ViT-B/16 student, the variant that unfreezes the last four transformer blocks (PTA) achieves the highest average accuracy (**65.36**%), especially for EuroSAT. Moderately expanding the fine-tuning window (PTA-FT8) retains most of the benefits, whereas full fine-tuning (PTA-FULL) improves most of the datasets but reduces robustness on EuroSAT, highlighting the usefulness of controlled adaptation depth.

Table 2 reports recall@{1,5} on Flickr30k and MS-COCO. The PTA variants again leverage the frozen ViT-B/32 teacher with a ViT-B/16 student. PTA-FT4 attains the best overall balance, improving COCO image-to-text R@1 to **25.24** (+1.3 over the strongest baseline) and text-to-image R@1 to **25.06**, while matching or exceeding baselines on Flickr30k. Increasing the number of trainable blocks (FT8/FULL) provides small additional gains on Flickr30k but slightly erodes COCO performance, reinforcing the observation that partial fine-tuning strikes the best stability–accuracy trade-off. Together with the classification metrics, these retrieval results demonstrate that PTA reliably transfers weak-model structure into the stronger encoder without sacrificing downstream generalisation.

**Robustness to Weak Model Quality**. A natural concern in weak-to-strong refinement is that the weak model must be relatively strong or well-calibrated for its similarities to be useful. To test how strongly PTA depends on the teacher, we conduct a controlled degradation study where we progressively corrupt the weak model (CLIP-B/32) at inference time using dropout rates $\{0.0, 0.1, 0.3, 0.5, 0.8\}$, while keeping the strong model architecture and all training hyperparameters fixed. Figure 5a plots the average zero-shot accuracy over seven downstream benchmarks for both the weak model and the PTA-trained strong model as a function of the dropout rate.

Table 1: Zero-shot classification accuracies (in %) on downstream datasets. The teacher model is a ViT-B/32 backbone, and the student model is ViT-B/16. All other baselines are discussed in Appendix A.2. FT4 and FT8 mean we only unfreeze the last 4 or 8 transformer blocks of the student, whereas FULL fine-tunes all vision blocks. Bold numbers mark the best entry per column.

| Method | CIFAR-10 | CIFAR-100 | Food-101 | ImageNet | DTD | Stanford Cars | EuroSAT | Avg. |
|---|---|---|---|---|---|---|---|---|
| SFT | 83.74 | 62.27 | **86.93** | 60.89 | 42.45 | 56.52 | 44.17 | 62.42 |
| Soft Align. | 89.39 | 66.35 | 81.98 | 63.41 | 43.83 | 58.50 | 40.27 | 63.39 |
| VL PseudoLabel | 87.23 | 64.19 | 79.69 | 62.63 | 43.83 | 57.17 | 38.55 | 61.90 |
| Fixed Threshold | 89.78 | 66.39 | 80.13 | 63.12 | 43.67 | 58.49 | 39.68 | 63.04 |
| Mom. Distill. | 89.83 | 66.36 | 80.47 | 63.33 | 43.88 | 58.97 | 38.39 | 63.03 |
| Aug. | 89.75 | 66.38 | 80.19 | 63.18 | 43.62 | 59.01 | 40.15 | 63.18 |
| PTA-FT4 | 89.74 | 65.76 | 82.36 | 64.14 | 44.68 | 60.50 | **50.36** | **65.36** |
| PTA-FT8 | 90.92 | 67.66 | 84.25 | 65.13 | **45.37** | 62.59 | 38.04 | 64.85 |
| PTA-FULL | **91.59** | **67.96** | 84.95 | **64.61** | 43.56 | **62.88** | 38.11 | 64.81 |

Table 2: Cross-modal retrieval on Flickr30k and MS-COCO (recall@1 and recall@5). All systems are fine-tuned on COCO train2014. FT4 and FT8 fine-tune only the last 4 or 8 vision blocks of the student, while FULL updates every block. Bold numbers highlight the best score in each column.

| | Flickr30k Dataset | | | | MS-COCO Dataset | | | |
|---|---|---|---|---|---|---|---|---|
| | Image to Text | | Text to Image | | Image to Text | | Text to Image | |
| Method | R@1 | R@5 | R@1 | R@5 | R@1 | R@5 | R@1 | R@5 |
| SFT | 81.00 | 95.60 | 79.70 | 95.30 | 22.69 | 41.69 | 21.94 | 41.29 |
| Soft Align. | 81.90 | 96.40 | 81.70 | 96.00 | 23.65 | 43.69 | 22.26 | 42.06 |
| VL PseudoLabel | 81.30 | 96.10 | 81.50 | 95.80 | 24.91 | 45.35 | 24.24 | 44.81 |
| Fixed Threshold | 83.00 | 96.20 | 81.30 | **96.40** | 23.91 | 43.72 | 23.84 | 43.69 |
| Mom. Distill. | 82.70 | 96.20 | 81.20 | 96.30 | 23.89 | 43.73 | 23.69 | 43.58 |
| Aug. | **83.40** | 96.20 | 81.40 | 96.30 | 23.97 | 43.67 | 23.85 | 43.72 |
| PTA-FT4 | 82.40 | **96.60** | **83.20** | 96.30 | **25.24** | **45.93** | **25.06** | **45.73** |
| PTA-FT8 | 82.20 | **96.60** | 81.60 | 96.30 | 24.97 | 45.66 | 24.71 | 45.27 |
| PTA-FULL | **83.40** | 96.20 | 81.10 | 96.00 | 24.09 | 43.88 | 23.75 | 43.48 |

As the dropout increases, the weak model collapses from 61.0% at dropout 0.0 to 3.54% at 0.8, a relative drop of 94.2%. In contrast, the strong model degrades only mildly: its accuracy decreases by less than one point under moderate corruption (dropout 0.3), and even at dropout 0.8 it still achieves 59.79%, a relative drop of only 8.9%.

**Data Efficiency: CC3M Scaling Study**. To evaluate whether PTA acts as a data-efficient refinement mechanism rather than merely adding computation, we conduct a scaling experiment on CC3M. We fine-tune a CLIP-B/16 student (last four ViT blocks trainable) for one epoch under identical hyperparameters, varying only the number of CC3M image–text pairs used: 1.0M, 2.0M, and 3.0M. All baselines and PTA use the same weak teacher (CLIP-B/32) and the same training setup; the only difference is how additional positive signals are extracted. Figure 5b shows the average accuracy over downstream benchmarks as a function of the number of CC3M pairs. The supervised baseline exhibits clear diminishing returns: increasing the dataset from 1.0M to 2.0M pairs yields a +2.91 point gain, whereas the next million (from 2.0M to 3.0M) adds only +1.40 points. On the same 3M examples, PTA improves performance by a further +1.25 points over the 3M supervised baseline, comparable to

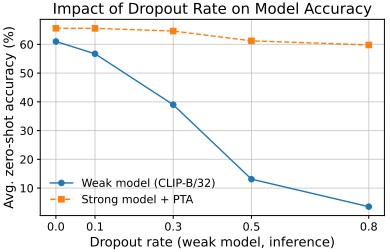

(a) Weak-model stress test

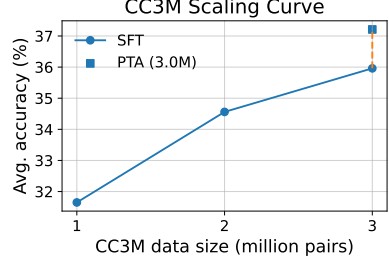

(b) CC3M scaling

Figure 5: (a) Robustness of PTA to weak-model corruption; (b) data-efficient gains on CC3M.

what standard fine-tuning would obtain by adding roughly 0.6M extra CC3M pairs. These results highlight that PTA yields non-trivial improvements even when the training data is fixed and relatively small. Rather than requiring additional multimodal pairs, PTA provides a statistically calibrated way to reuse in-batch negatives as auxiliary positives, improving alignment in a data-efficient manner.

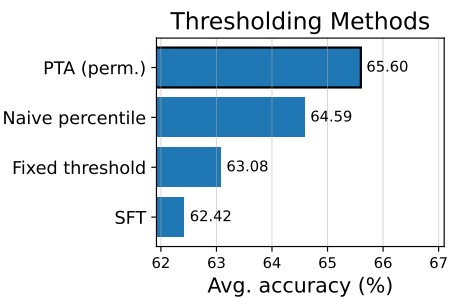

Figure 6: Thresholding ablation on the seven-dataset average..

**Ablation: Thresholding Strategies.** PTA relies on a calibrated rule for deciding which in-batch pairs should be treated as additional positives based on weak similarities. Here we compare several thresholding strategies under an identical weak-to-strong setting (same weak teacher, backbone, optimizer, and data), and report their seven-dataset average zero-shot accuracy (cf. Table **??**). The variants are: (i) *SFT.* Standard supervised fine-tuning on COCO without any weak-similarity filtering, (ii) *Fixed threshold.* A single global cosine-similarity cutoff is tuned on COCO validation and reused for all batches, (iii) *Naive percentile.* The cutoff is set to the 99th percentile of weak similarities on ground-truth COCO pairs and kept fixed during training, and (iv) *PTA (permutation).* Our method derives the cutoff from the permuted null distribution of weak similarities at a chosen significance level $\alpha$, yielding a dataset- and teacher-adaptive threshold with explicit control over the tolerated false-alarm rate. Figure 6 summarizes the seven-dataset averages for these strategies. Both fixed threshold and naive percentile improve over the SFT baseline, showing that even simple heuristics can benefit from weak similarities. However, the permutation-based PTA threshold achieves the highest accuracy, indicating that calibrating against a data-driven null distribution yields cleaner auxiliary positives than fixed global cutoffs.

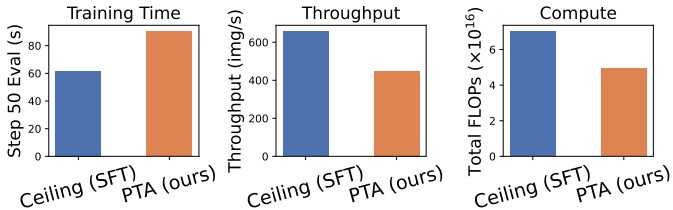

Figure 7: Training efficiency on COCO (CLIP-B/32). PTA incurs $\sim$1.3–1.4$\times$ slowdown per step due to the weak model forward, consistent across early steps.

**Training Efficiency Comparison.** We compare the training efficiency of PTA against a ceiling baseline (standard supervised fine-tuning without a weak model). Figure 7 summarises throughput and overhead. On the first evaluation trigger (step 50), the PTA run required 90.2s per evaluation at 448.9 samples/s, versus 61.7s at 656.4 samples/s for the ceiling baseline—a $\sim$1.4$\times$ slowdown.

At step 100, the pattern persisted: PTA achieved 74.2 images/s compared to 54.8 images/s for the ceiling, reflecting a $\sim$35% per-step overhead that closely mirrors the cost of one additional CLIP forward pass (the weak model). Overall, PTA's weak-model inference effectively adds one extra CLIP forward per batch, yielding a consistent 1.3–1.4$\times$ slowdown relative to SFT. Additional costs arise from log book-keeping and duplicate forward passes, but these remain minor compared to the weak model overhead. Because PTA jobs often early-stop when convergence criteria are met, the total recorded FLOPs may appear lower, though if both methods train to the full schedule, the GPU-hours scale linearly with the per-step slowdown.

**Additional experiments.** We includes more experiments in Appendix A.5.

## 5 CONCLUSION

In this paper, we present a novel framework for enhancing the image-text alignment of a strong model through the supervision provided by a weak model. Our method addresses an important challenge in multimodal learning: how to effectively leverage noisy supervision signals to improve model performance. The key innovation lies in our two-stage approach: first, a threshold-based refinement mechanism that automatically filters unreliable supervision signals by analysing the inherent structure of the dataset, and second, an enhanced supervision strategy that utilises continuous similarity scores to provide richer training signals. Empirical results on multiple benchmark datasets validate the effectiveness of our method, showing consistent improvements over existing approaches.

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

Table 3: COCO-2014 dataset statistics.

| Split | Images | Object categories |
|---|---|---|
| Train2014 | 82,783 | 80 |
| Val2014 | 40,504 | |

Table 4: Evaluation datasets. Retrieval corpora list the number of captions per image; classification corpora list the number of labels.

| Dataset | Task | Images | Captions / labels |
|---|---|---|---|
| COCO val2014 | Retrieval | 40,504 | 5 captions |
| Flickr30k test | Retrieval | 1,000 | 5 captions |
| ImageNet val | Classification | 50,000 | 1,000 labels |
| Stanford Cars test | Classification | 8,041 | 196 labels |
| DTD test | Classification | 1,880 | 47 labels |
| EuroSAT val | Classification | 27,000 | 10 labels |
| CIFAR-100 test | Classification | 10,000 | 100 labels |
| CIFAR-10 test | Classification | 10,000 | 10 labels |
| Food-101 val | Classification | 25,250 | 101 labels |

## A  APPENDIX

### A.1  DATASETS

Our experiments leverage the Microsoft COCO (Lin et al., 2014)(Common Objects in Context) 2014 dataset (statistics are shown in Table 3) , which is a large-scale benchmark for object detection, segmentation, keypoint detection, and captioning. It contains images across 80 object categories, split into 82,783 training images and 40,504 validation images. Each image is annotated with object class labels and bounding boxes, and many are paired with natural language captions. COCO 2014 has become a standard resource for evaluating visual recognition and image–text alignment methods.. To evaluate generalisation, we test on object recognition tasks spanning CIFAR-10 (Krizhevsky et al., 2009), CIFAR-100 (Krizhevsky et al., 2009), Food-101 (Bossard et al., 2014), ImageNet (Deng et al., 2009), Describable Textures (DTD) (Cimpoi et al., 2014), StanfordCars, and EuroSAT (Helber et al., 2019), which cover diverse domains (e.g., natural scenes, textures, satellite imagery). For cross-modal retrieval, we use Flickr30K (Plummer et al., 2015) and MS-COCO (Lin et al., 2014), standard benchmarks for image-text matching. This dual training-testing framework validates robustness to both noisy and curated supervision while measuring downstream task adaptability across modalities. Evaluation dataset statistics are shown in Table 4.

### A.2  BASELINES

We compare our method against a comprehensive set of baselines: (1) **CLIP supervised fine-tuning (SFT) baselines.** We include standard CLIP variants: CLIP-B16 in their supervised fine-tuning. These provide reference points for direct adaptation. (2) **Weak-to-Strong Augmentation (Aug.).** Proposed by Burns et al. (2023), AUG incrementally augments positives discovered by weaker models in multi-stage training. This is the closest prior to our weak-to-strong framework. (3) **Soft Align.** Following recent work on soft-label alignment (Huang et al., 2024), we use similarity scores from a weaker model to generate soft positive labels, replacing the binary supervision in the contrastive loss. (4) **VL PseudoLabel.** Inspired by multi-label pseudo-labeling approaches for vision-language models (Xing et al., 2024), this baseline allows each image to have multiple captions marked positive if predicted above a confidence threshold. (5) **Fixed Threshold.** As a naive baseline, we apply a constant threshold (e.g., 0.3) on weak model similarity scores to decide additional positives (Fang et al., 2023). This tests whether our permutation-based statistical control is necessary. (6) **Momentum Distillation (Mom. Distill.).** Following ALBEF (Li et al., 2021), we maintain a momentum-updated copy of the weak model as a teacher, and minimize KL divergence between the teacher's and student's similarity distributions.

## A.3 IMPLEMENTATION DETAILS

We implement our training pipeline in PyTorch using the Hugging Face `transformers` stack. Random seeds for Python, NumPy, and PyTorch are fixed to ensure deterministic data shuffling and initialisation. Unless noted otherwise, all experiments fine-tune the strong model on a CoCo dataset.

**Models.** The weak teacher is a frozen CLIP ViT-B/32 encoder that produces image–text similarity logits. The strong learner is CLIP ViT-B/16 with a trainable vision tower and projection heads; its text encoder remains frozen. Both models are instantiated with the default CLIP image preprocessing (bicubic resize, center crop, RGB conversion, mean–std normalisation).

**Threshold calibration.** Before training, we estimate a weak-model threshold $Q$ that separates reliable positives from noise. We repeatedly sample minibatches from COCO train2014, obtain teacher similarities for the matched pairs, and record scores again after randomly permuting captions within the batch. The permuted scores form the null distribution $\mathcal{P}_{\text{null}}$, and we select $Q \leftarrow (1 - \alpha)$-th percentile of $\mathcal{P}_{\text{null}}$. To determine a stable sampling budget, we varied both the number of pairs and the number of negatives per image: sample sizes of 100, 1,000, and 5,000 pairs each for five iterations. We observed that 1,000 pairs combined with five iterations (5,000 total pairs) converged to the same percentile as the larger setting, while being significantly cheaper. This configuration is therefore used for every experiment; the resulting $Q$ is cached and reused throughout training, and the entire calibration is rerun for each seed.

**Training recipe.** We optimize the stronger model with AdamW (learning rate $\eta = 10^{-6}$, weight decay $10^{-4}$, cosine decay schedule, 10% warmup) for $E = 20$ epochs and batch size $B = 4096$. Gradient checkpointing, BF16 mixed precision, and gradient clipping ($\|\nabla\|_2 \leq 1$) are enabled. A calibration routine computes the null score distribution $\mathcal{P}_{\text{null}}$ by evaluating $f$ on randomly permuted image–caption pairs; the 99th percentile serves as the PTA threshold $Q = 0.2645$ and is cached per run. Unless otherwise stated, the loss combines symmetric InfoNCE and our permutation-thresholded alignment objective in a $0.5/0.5$ ratio. The weak teacher is always kept in eval mode with gradients disabled. We used 10% data as a validation set for early stopping.

**Implementation of fine-tuning a few layers** We control adaptation depth through the `finetune_vision_last_n` flag: before training the strong CLIP encoder, we freeze every parameter in the vision tower and then unfreeze only the last $n$ transformer blocks (and always keep the visual projection head trainable). Passing $n = 4$ ("FT4") therefore updates just the top four blocks, $n = 8$ ("FT8") unlocks the top eight, and setting $n = -1$ (our default "FULL") exceeds the backbone length and effectively fine-tunes all blocks. The text encoder stays frozen in every configuration. Note here, we initialise those unfrozen layers from the original CLIP, where we empirically random initialisation cannot perform well in our setting, as we have a limited dataset and computes to train from scratch.

**Evaluation.** After training, we freeze the strong encoder and compute (i) zero-shot accuracy on the standard validation splits of ImageNet, Stanford Cars, DTD, EuroSAT, CIFAR-100, CIFAR-10, and Food-101 using cosine similarity between image features and prompt embeddings "*A photo of a {label}*"; and (ii) image-to-text / text-to-image recall@$\{1,5\}$ on COCO val2014 and Flickr30k. Averaged metrics across the validation sets are also reported.

**Hardware and runtime.** All experiments run on NVIDIA H100 GPUs with $80\,\text{GB}$ memory. Unless noted otherwise, we use a single GPU, batch size 4096, and 4 dataloader workers. Fine-tuning the baseline (no teacher) completes in approximately 5.8 GPU-hours per run; enabling our PTA loss adds a frozen weak-model forward pass and extends wall-clock time to roughly 7.5 GPU-hours (about $1.3\times$ slower). No distributed data parallelism is required at this scale.

## A.4 RELATED WORK

**Vision-language models** Since the introduction of CLIP by Radford et al. (2021), numerous models building upon its multimodal representation learning framework have emerged. Subsequent work has explored architectural improvements to CLIP, such as ALIGN (Jia et al., 2021) which scales

up the training data and model size, and Florence (Yuan et al., 2021) which incorporates additional visual reasoning capabilities. Recent models like BLIP (Li et al., 2022) and LiT (Zhai et al., 2022) have extended CLIP's contrastive pre-training approach to incorporate additional modalities like text and layout. The success of CLIP has firmly established the effectiveness of large-scale contrastive learning on aligned image-text pairs for learning transferable visual representations.

**CLIP fine-tuning** Recent advancements in fine-tuning CLIP models have led to various parameter-efficient methods aimed at enhancing performance on downstream tasks. The Context Optimization (CoOp) (Zhou et al., 2022b) approach introduces learnable text prompts, enabling the model to adapt to specific tasks by optimizing these prompts during training. Building upon this, Conditional Prompt Learning (CoCoOp) (Zhou et al., 2022a) extends CoOp by conditioning the prompts on image instances, thereby improving generalization across diverse visual concepts. Alternatively, CLIP-Adapter (Gao et al., 2024a) incorporates lightweight residual-style adapters into CLIP's architecture, facilitating the learning of task-specific features without extensive retraining. Tip-Adapter (Zhang et al., 2021) further streamlines this process by constructing a key-value cache model from the few-shot training set, enabling efficient adaptation with minimal or no training. Additionally, methods like Visual Prompt Tuning (VPT) (Jia et al., 2022) add learnable prompt tokens in the input space, fine-tuning the model effectively for various tasks. These strategies collectively aim to enhance CLIP's adaptability and performance while maintaining computational efficiency. Our method differs from previous approaches since we address a distinct problem setting (Image-text alignment).

**Bootstrapping and Pseudo-Labelling.** Several works attempt to mitigate noise in image-text alignment through bootstrapping or pseudo-labelling. BLIP (Li et al., 2022) employs caption bootstrapping and multi-stage training to refine noisy web-crawled supervision. More recent studies propose pseudo-labelling strategies to relax one-hot supervision, such as Cross-Modal Similarity Regulation (Jiang et al., 2023), multi-label pseudo-labelling (Xing et al., 2024), and soft-label alignment (Huang et al., 2024). These methods highlight the importance of leveraging softer or multiple alignment signals. Our approach shares this motivation but differs by introducing a statistically grounded mechanism (permutation testing) to identify semantically meaningful extra positives with controlled false alarm rates.

**Knowledge Distillation and Weak-to-Strong Supervision.** Knowledge distillation techniques, such as ALBEF (Li et al., 2021), employ momentum encoders to refine alignment signals from auxiliary models. Similarly, weak-to-strong transfer has been investigated in broader contexts (Burns et al., 2023), emphasising the role of weaker models in guiding stronger models. PTA builds upon this weak-to-strong philosophy but departs from direct distillation: instead of transferring continuous similarity scores, it adaptively thresholds weak-model signals using a permutation-derived null distribution, producing statistically validated supervision.

**Mitigating False Negatives.** False negatives—i.e. unpaired image–text instances that are semantically related—have recently received direct attention in vision–language pretraining. For instance, MAFA (Byun et al., 2024) proposes an efficient connection mining process to convert some hard false negatives into positives, and smooths the contrastive ITC loss to reduce over-penalisation. Empirically, MAFA shows that doing this yields substantial downstream gains, especially under hard-negative sampling strategies. Our approach (PTA) differs in that we (i) use a permutation-based null distribution to provide statistical thresholds with error control; (ii) treat all non-paired but high-score items, above that threshold, as additional positives (weak-to-strong supervision), rather than only removing or heuristically filtering negatives; and (iii) validate prevalence via multi-caption datasets, cross-model agreement, and human audit rather than relying solely on mining or smoothing heuristics.

Several works seek to mitigate noisy supervision in contrastive multimodal learning. Methods such as PyramidCLIP (Gao et al., 2022), SoftCLIP (Gao et al., 2024b), and CLIP-PSD (Andonian et al., 2022) adjust the contrastive objective or logit geometry to soften the effect of incorrect negatives or low-quality positives during training, but operate in the standard in-batch regime without explicit statistical control. AdaCL (Pan et al., 2025) focuses on *clone negatives* captions that are semantically close but not exact matches—by adaptively reshaping the positive logit, but still treats all non-matching pairs as negatives. A separate line of work (Han et al., 2025) addresses *noisy correspondences* in large-scale pre-training corpora, where some annotated positive pairs are mismatched and must be repaired or reweighted; these methods assume access to the full pre-training dataset

Table 5: **Weak–Strong Gap Experiment (B/32 → L/14).** PTA achieves consistent improvements over the ceiling baseline across multiple datasets.

| Method | INet | Cars | DTD | EuroSAT | CIFAR100 | CIFAR10 | Food101 | Avg |
|---|---|---|---|---|---|---|---|---|
| SFT | 70.06 | 73.21 | 52.71 | **59.80** | 76.63 | **95.60** | 88.12 | 73.73 |
| PTA (ours) | **71.03** | **75.29** | **53.35** | 57.99 | **77.24** | 95.49 | **89.38** | **74.25** |

Table 6: **Null provider ablation (B/32 vs. B/16).** Both weak-model choices lead to nearly identical performance when supervising CLIP-L/14, indicating robustness of the PTA permutation test.

| Null Provider | INet | Cars | DTD | EuroSAT | CIFAR100 | CIFAR10 | Food101 | Avg |
|---|---|---|---|---|---|---|---|---|
| B/32 (ours) | 64.14 | **60.50** | 44.68 | 50.36 | **65.76** | **89.74** | **82.36** | 65.36 |
| B/16 (alt) | 64.14 | 60.37 | 44.68 | **51.25** | 65.73 | 89.54 | 82.31 | **65.43** |

and focus on cleaning noisy positives. Our setting differs from all of the above: we study post-hoc weak-to-strong refinement on a small, clean downstream dataset, where positives are correct but the contrastive loss introduces hard false negatives through in-batch sampling.

## A.5 ADDITIONAL EXPERIMENTS

### A.5.1 EFFECT OF WEAK–STRONG MODEL GAP.

To test whether PTA remains effective when scaling from a relatively weak teacher to a stronger student, we conducted experiments using CLIP-B/32 as the weak model and CLIP-L/14 as the strong model. We compared against a ceiling baseline where the strong model is directly fine-tuned without PTA. Table 5 summarises the results across seven transfer benchmarks. PTA consistently improves over the ceiling baseline. On ImageNet validation, PTA reaches 71.0% versus 70.1% for the ceiling. Gains are more pronounced on fine-grained datasets such as Stanford Cars (+2.1%) and Food101 (+1.3%), and PTA also improves average accuracy from 73.7% to 74.3%. These results confirm that PTA's permutation-based filtering continues to be beneficial even when the gap between weak and strong models is large (B/13 → L/14).

### A.5.2 NULL DISTRIBUTION FROM WEAK VS. STRONG MODELS.

PTA is designed to build the permutation null distribution from a weaker model, under the assumption that weak similarities are more diffuse and less biased toward high-confidence matches. To test whether using the strong model itself could serve as the null generator, we compared two variants on COCO fine-tuning: (1) PTA with CLIP-B/32 as the weak model (our default), and (2) PTA with CLIP-B/16 as the null provider for the CLIP-L/14 strong model. Table 6 reports transfer accuracies across seven benchmarks. We find that both variants yield almost identical results: average accuracy improves marginally from 65.36% (B/32 as weak) to 65.43% (B/16 as weak). Dataset-level differences are within noise (e.g., EuroSAT 50.36 vs 51.25, Stanford Cars 60.50 vs 60.37). This suggests that PTA is robust to the choice of null provider: using a stronger teacher does not meaningfully alter downstream transfer performance, likely because the permutation thresholding adapts to the null distribution scale.

### A.5.3 ABLATION ON THE LOSS BALANCING COEFFICIENT $\lambda$

Our framework combines two loss components: the weighted weak supervision term and the standard contrastive loss. The hyperparameter $\lambda$ controls the balance between them. We vary $\lambda$ to study its influence on transfer performance after fine-tuning on COCO (train2014). Results are summarised in Table 9. We observe that performance is stable across a wide range of $\lambda$. In particular, $\lambda = 0.50$ and $\lambda = 0.75$ achieve the best average accuracies (65.36% and 65.49%, respectively), indicating that a balanced or slightly higher emphasis on the weak supervision component is most beneficial. Extremely skewed values ($\lambda = 0.10$ or $0.90$) slightly reduce performance, suggesting that both losses are complementary and should be jointly optimised.

### A.5.4 ABLATION ON THE CHOICE OF SIGNIFICANCE LEVEL $\alpha$

Table 7: Zero-shot classification accuracy (%) of CLIP-B/32 with different dropout rates during inference. Higher dropout rates simulate progressively weaker models.

| Dropout | Caltech101 | CIFAR10 | CIFAR100 | DTD | Flowers102 | Food101 | SUN397 | ImageNet | Average |
|---|---|---|---|---|---|---|---|---|---|
| 0.0 | 59.58 | 58.38 | 42.66 | 36.52 | 60.60 | 88.71 | 80.55 | 61.00 | 61.00 |
| 0.1 | 52.77 | 50.22 | 36.06 | 39.74 | 51.66 | 89.26 | 77.36 | 56.72 | 56.72 |
| 0.3 | 28.31 | 21.74 | 26.17 | 26.36 | 38.93 | 81.08 | 50.43 | 39.00 | 39.00 |
| 0.5 | 7.32 | 4.44 | 10.96 | 12.51 | 10.02 | 33.37 | 13.09 | 13.10 | 13.10 |
| 0.8 | 0.11 | 0.50 | 2.07 | 9.77 | 1.06 | 10.32 | 0.97 | 3.54 | 3.54 |

Table 8: Classification accuracy (%) after fine-tuning CLIP-B/16 using PTA with CLIP-B/32 weak models at different dropout rates. All models are fine-tuned on COCO Captions with identical hyperparameters.

| Weak Dropout | Caltech101 | CIFAR10 | CIFAR100 | DTD | Flowers102 | Food101 | SUN397 | ImageNet | Average |
|---|---|---|---|---|---|---|---|---|---|
| 0.0 | 64.14 | 60.40 | 44.57 | 50.51 | 66.45 | 89.38 | 83.77 | 65.60 | 65.60 |
| 0.1 | 64.13 | 59.47 | 44.52 | 50.94 | 66.58 | 89.36 | 83.98 | 65.57 | 65.57 |
| 0.3 | 62.80 | 56.06 | 44.31 | 51.17 | 65.56 | 89.47 | 83.04 | 64.63 | 64.63 |
| 0.5 | 58.74 | 47.10 | 44.73 | 49.75 | 62.43 | 88.72 | 77.12 | 61.23 | 61.23 |
| 0.8 | 56.84 | 47.78 | 40.53 | 46.51 | 60.81 | 87.74 | 78.30 | 59.79 | 59.79 |

Our method introduces a significance level $\alpha$ to control the selection of plausible negatives. To assess its effect, we vary $\alpha$ and evaluate transfer performance on seven downstream benchmarks. Results are reported in Table 10. We find that the best overall performance is obtained around $\alpha = 0.01$, which achieves the highest average accuracy of 65.36%. Smaller choices of $\alpha$ (e.g., 0.1, 0.05) perform comparably well, while overly large or overly conservative values (e.g., $\alpha = 0.001$) slightly reduce accuracy. These results suggest that while our method is robust to a broad range of $\alpha$, setting $\alpha$ near the default training value offers a good balance across tasks.

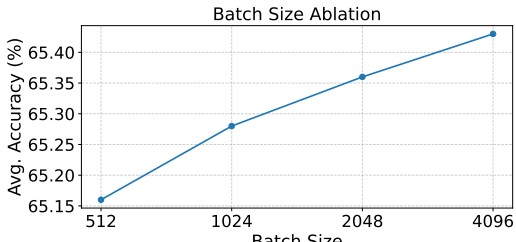

Figure 8: Average accuracy (%) across seven downstream datasets after fine-tuning on COCO (train2014) with PTA.

### A.5.5 ABLATION ON BATCH SIZE.

Results are reported in Figure 8. To examine the effect of batch size on transfer performance, we fine-tune our model on COCO using the proposed PTA objective with $alpha = 0.01$ and evaluate on a diverse set of downstream benchmarks. Overall, performance remains stable across batch sizes ranging from $B = 512$ to $B = 4096$. Average accuracy fluctuates within a narrow band (65.16–65.43), suggesting that our method is robust to batch size, despite the prevalence of false negatives increasing at larger $B$ (see Figure 2). This indicates that PTA effectively mitigates the negative impact of semantically related in-batch samples.

### A.6 VISUALISATION

### A.6.1 QUALITATIVE RETRIEVAL IMPROVEMENTS

To illustrate the effect of our method beyond quantitative metrics, we provide qualitative retrieval examples comparing Soft Align with PTA. As shown in the retrieval visualizations in 9, PTA consistently ranks semantically correct images higher than Soft Align. In several cases, Soft Align retrieves visually similar but semantically mismatched distractors (e.g., motorcycles without a teddy bear, or generic train-track scenes), whereas PTA elevates the ground-truth or more semantically faithful matches to the top positions. These examples highlight PTA's ability to better distinguish fine-grained semantic cues even under visually cluttered or ambiguous contexts.

Table 9: **Ablation on the loss balancing coefficient** $\lambda$. Accuracy (%) on seven downstream datasets after fine-tuning on COCO (train2014) with different values of $\lambda$. Avg. is the mean over all tasks.

| $\lambda$ | ImageNet | Cars | DTD | EuroSAT | CIFAR100 | CIFAR10 | Food101 | Avg. |
|---|---|---|---|---|---|---|---|---|
| 0.10 | 63.65 | 59.58 | 44.47 | 51.47 | 65.77 | 89.46 | 80.97 | 65.05 |
| 0.25 | 63.89 | 60.02 | 44.36 | 51.60 | 65.67 | 89.59 | 81.47 | 65.23 |
| 0.50 | 64.14 | 60.50 | 44.68 | 50.36 | 65.76 | 89.74 | 82.36 | 65.36 |
| 0.75 | 63.61 | 60.04 | 44.63 | 52.03 | 65.51 | 90.20 | 82.40 | **65.49** |
| 0.90 | 64.46 | 60.48 | 45.11 | 44.35 | 66.27 | 89.42 | 83.70 | 64.83 |

Table 10: **Ablation on significance level** $\alpha$. Accuracy (%) on seven downstream datasets after fine-tuning on COCO (train2014) with different $\alpha$ values. Avg. is the mean over all tasks.

| $\alpha$ | ImageNet | Cars | DTD | EuroSAT | CIFAR100 | CIFAR10 | Food101 | Avg. |
|---|---|---|---|---|---|---|---|---|
| 0.1 | 63.61 | 59.63 | 43.99 | 51.45 | 65.88 | 89.45 | 80.85 | 64.98 |
| 0.05 | 63.90 | 60.22 | 44.41 | 51.38 | 65.77 | 89.36 | 81.65 | 65.24 |
| 0.01 | 64.14 | 60.50 | 44.68 | 50.36 | 65.76 | 89.74 | 82.36 | **65.36** |
| 0.001 | 63.13 | 59.32 | 44.15 | 45.47 | 65.60 | 90.15 | 81.01 | 64.12 |

### A.6.2   GRAD-CAM

We further analyze how PTA modifies the model's visual focus by comparing Grad-CAM heatmaps (Selvaraju et al., 2016). As shown in 10, PTA results in noticeably sharper and more semantically aligned attention maps across diverse query categories. For instance, in the airplane example, Soft Align spreads attention broadly across background regions, while PTA concentrates on the aircraft body and windows. For sheep and motorcycle queries, PTA improves focus on the primary foreground objects and suppresses irrelevant background clutter.

Query image — dog

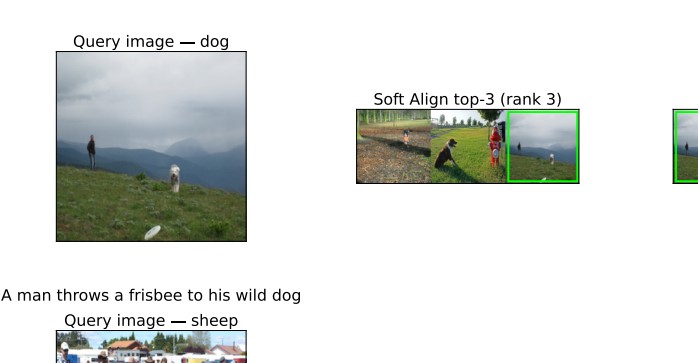

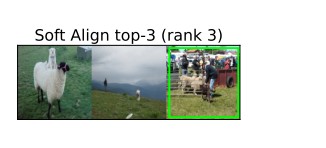 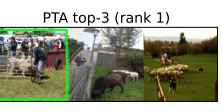

A man throws a frisbee to his wild dog

Query image — sheep

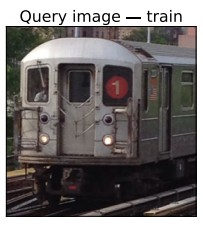 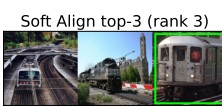 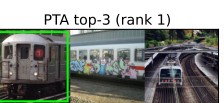

A man standing next to a dog near a sheep penn.

Query image — train

A silver train sitting on top of train tracks.

Figure 9: Qualitative retrieval comparison between Soft Align and PTA.

## B REPRODUCIBILITY STATEMENT

Code, training scripts, and evaluation utilities will be publicly released upon publication. We include implementation details in Appendix A.3 .

## C AI USAGE CLARIFICATION

Large Language Models were employed solely to enhance grammar and readability. All aspects of research design, analysis, and interpretation were conducted by the authors.

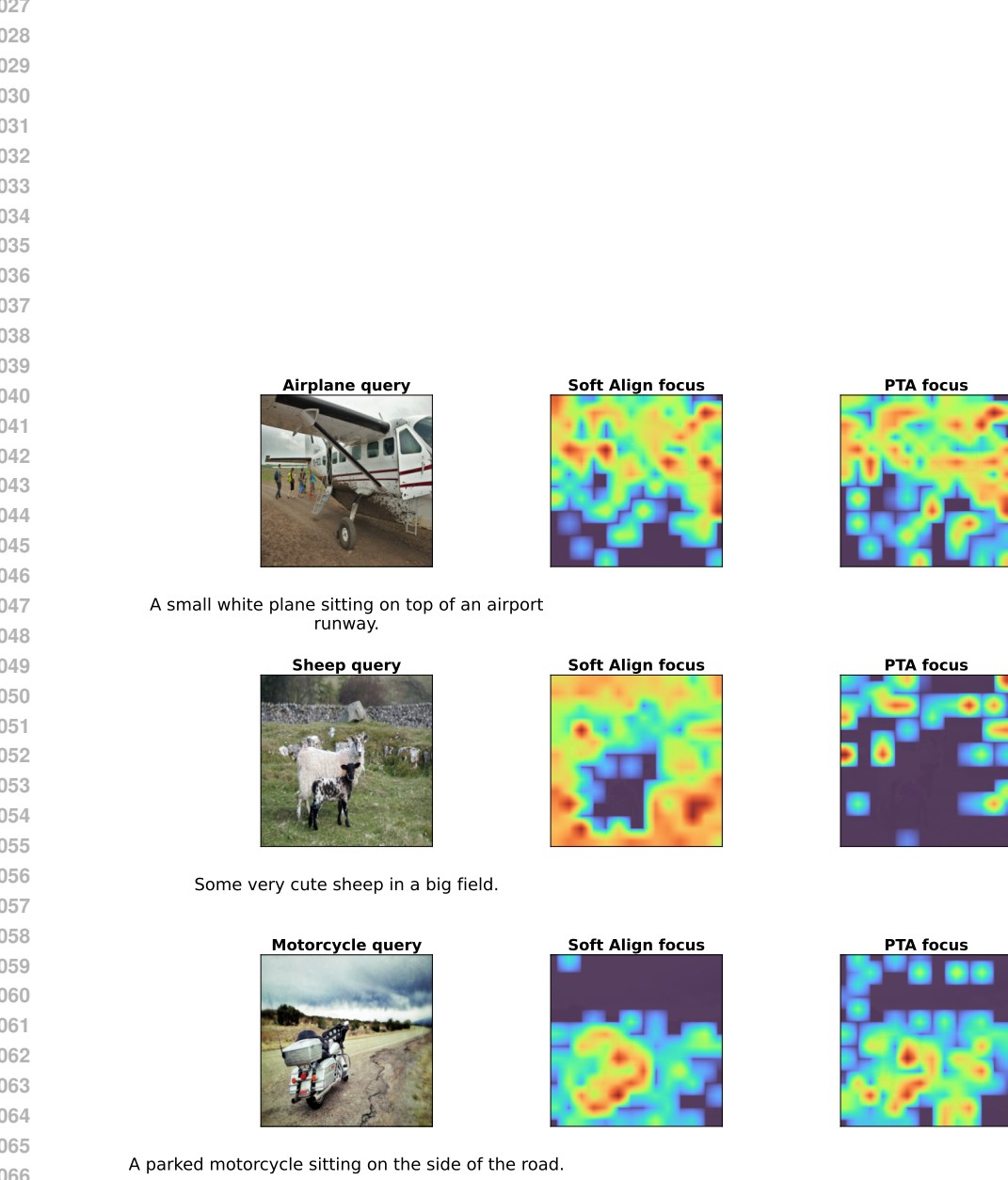

Figure 10: Comparison of visual grounding via Grad-CAM before and after PTA.

