# OpenReview forum: "Beyond Hard Supervised Fine-tuning: Enhancing Image-text Alignment of Strong Models with Weak Models"
_ICLR.cc/2026/Conference — Submitted to ICLR 2026_

### Official Review · Reviewer_zZyZ · 2025-10-15

**Soundness:** 2
**Presentation:** 3
**Contribution:** 3
**Rating:** 4
**Confidence:** 5

**Summary:**

This paper addresses the problem of "false negatives" in CLIP. In standard training, all unpaired image-text combinations in a batch are treated as negatives, but many are semantically related, leading to noisy supervision that hinders model performance. The authors propose a "weak-to-strong" supervision framework. PTA uses a weaker model to guide the fine-tuning of a stronger model, with the process of Permute-then-Adapt. The method is evaluated on zero-shot classification, and two cross-modal retrieval tasks, demonstrating improvements over baselines.

**Strengths:**

- I think the idea of using permutation test to solve FP problem is novel. This is a improvement of traditional hard supervised fine-tuning. Moving beyond fixed, heuristic thresholds to a statistically adaptive one is a significant conceptual leap. The "weak-to-strong" framing is aptly applied to the multimodal domain.

- Strong empirical study: The paper provides thorough analysis of false negatives in CLIP, which is quite solid.

- Good writing. The paper is quite easy to follow, which contains essential background for CLIP.

**Weaknesses:**

- Limited literature review: There exists quite a few works tackling false negatives in CLIP, including but not limited to [1], [2], [3]. More baselines should be included in your literature review.

- Limited baseline comparison. As mentioned above, more baselines should be evaluated to improve the completeness of this work.

- The pre-requisite for this method is laborious. You must shuffle datasets for many iterations before obtaining the distribution.

- The choice of $\alpha$ really matters in scalable contrastive learning. While the authors simply evaluate on COCO and F30K, more diverse datasets should be included to prove the generalization ability.

- Computational overhead: While the authors note a ~1.3x slowdown (Figure 6), this is a non-trivial cost, especially when considering large-scale pre-training. The requirement to compute the weak model's similarity for all in-batch negative pairs for every training step (to compute L_PTA_batch) adds a persistent overhead. A more detailed discussion on the scalability of this approach and potential optimizations (e.g., caching, using a much smaller proxy model as the "weak" model) would be beneficial.


- Sensitivity to weak model quality: The method's success hinges on the weak model providing a meaningful similarity distribution. The ablation in A.5.2 shows robustness between B/32 and B/16, but what happens if the weak model is very weak or poorly calibrated? A systematic analysis of how the performance gap and quality of the weak model affect the strong model's final performance would provide valuable insights into the limits of this approach.

- Clarity on loss formulation: The loss functions in Equations (12) and (13) could be clarified. The denominator sums over all B items in the batch, but the numerator only sums over those where l_km^PTA = 1. This seems to encourage the model to assign high probability to any of the soft positives, which is a form of multiple-positive learning. This is a valid approach, but the connection to and difference from other multiple-positive losses could be discussed more explicitly.



[1]. Mitigating Noisy Correspondence by Geometrical Structure Consistency Learning, CVPR 2024.


[2]. Discovering Clone Negatives via Adaptive Contrastive Learning for Image-text Matching, ICLR 2025.


[3]. Unlearning the Noisy Correspondence Makes CLIP More Robust, ICCV 2025.

**Questions:**

As mentioned in `Weaknesses`. I am looking forward to the authors' rebuttal, experiments, and the updated manuscript.

---

> ### Author Response · Authors · 2025-11-22
>
> We thank Reviewer for recognising the permutation-test based treatment of false positives and our move beyond fixed heuristic thresholds as a novel and conceptually meaningful contribution of the weak-to-strong PTA framework. We will address the following specific concerns.
>
> **Weaknesses:**
>
> > w1-1: Limited literature review: There exists quite a few works tackling false negatives in CLIP, including but not limited to \[1\], \[2\], \[3\]. More baselines should be included in your literature review.
> > w1-2: Limited baseline comparison. As mentioned above, more baselines should be evaluated to improve the completeness of this work.
>
> **q-w1:** We thank the reviewer for pointing out additional interesting works on mitigating noisy correspondences [1–3]. We will expand the related work section to explicitly discuss these methods and clarify the relation to PTA. In our work, we study **weak-to-strong refinement** in a *post-hoc* fine-tuning setting, and our key contribution is a **permutation-based null distribution** that provides statistically calibrated thresholds for selecting additional positives from a weak model. We have carefully reviewed the three works mentioned by the reviewer and would like to clarify the differences as follows:
>
> - [1] tackles **noisy correspondence learning**, where some annotated image–text *matches* are actually mismatched or only weakly related, and uses cross- and intra-modal geometry to assign soft correctness scores and reweight these *positive* pairs during training. In contrast, our PTA framework assumes clean positives and instead focuses on **false negatives** introduced by in-batch sampling in CLIP-style contrastive learning, using a permutation-based null similarity distribution and a weak teacher to detect and relax unreliable *negatives* when adapting a stronger model. It is reasonable to treat it as related but not necessary a baseline.
>
> - [3] also studies noisy image–text pairs in CLIP, but it operates in a very different setting from our work. NCU assumes access to the original large-scale pretraining corpus (e.g., CC3M/CC12M/YFCC) and uses optimal transport with learned hardest-negative prompts to *unlearn* low-confidence pairs and repair the pretraining. In contrast, our method works as a lightweight weak-to-strong fine-tuning scheme on a much smaller post-training dataset, without requiring pretraining data, and handles false negatives via permutation-based, statistically calibrated thresholds on similarity scores rather than via machine unlearning.
>
> - AdaCL [2] focuses on *clone negatives* as sub-optimal captions compared to ground-truth in supervised image–text matching and addresses them by adaptively rescaling the positive logit within a fixed-label contrastive loss. Different from PTA, they treat those examples still as negative but scale/shift of the positive logit to reduce their impact. It is different how we treat false negatives as *soft positives*, But it is interesting to compare.
>
> We adapt AdaCL to the vision–language fine-tuning setting by applying its adaptive margin mechanism to the image–text contrastive loss. Following table reports the performance of PTA and AdaCL on the fine-tuning setup, evaluated seven vision benchmarks. While AdaCL attains a reasonable average accuracy of 56.86%, PTA substantially outperforms it with 65.36% (+8.5%).
>
> We attribute this performance gap to a fundamental difference in objectives. AdaCL infers sample reliability purely from the student’s own embedding distribution, which is unstable under the rapid distribution shifts induced by fine-tuning. In contrast, PTA explicitly leverages the weak teacher’s confidence to filter out noisy pairs. Moreover, AdaCL’s soft margin adjustment still forces the model to learn from all samples, whereas PTA’s hard thresholding prevents the strong model from overfitting to teacher errors or noisy data, leading to stronger generalization.
>
> | Method                             | Average Accuracy (%) |
> | ---------------------------------- | -------------------- |
> | AdaCL (Adapted)                    | 56.86                |
> | **Ours (PTA)**                     | **65.36**            |
>
> In conclusion, all the above methods does not consider a small or weak model to identify false negatives under explicit type-I error control, which is the main novelty of our PTA framework. **We study how we can identify the extra information from negatives via a small/weak model.**

---

> > ### Author Response · Authors · 2025-11-22
> >
> > > w2: The pre-requisite for this method is laborious. You must shuffle datasets for many iterations before obtaining the distribution.
> >
> > **q-w2:** The permutation step is *not* performed over the entire dataset or at every iteration. As detailed in Implementation Details (Appendix A.3), we estimate the null distribution from a small subsample (1k pairs × 5 iterations ≈ 5k pairs), compute the (1−$\alpha$)-percentile threshold once, and cache it for the entire training run (also used for another fine-tuning datasets). This calibration takes a negligible fraction of the total runtime (minutes vs. hours for fine-tuning) and does not materially increase overall cost.
> >
> > > w3: The choice of $\alpha$ really matters in scalable contrastive learning. While the authors simply evaluate on COCO and F30K, more diverse datasets should be included to prove the generalization ability.
> >
> > **q-w3:** We agree that $\alpha$ is an important hyper-parameter in scalable contrastive learning. Our ablation in Table 3 varies $\alpha$ over {0.1, 0.05, 0.01, 0.001} and fine-tunes on COCO, then evaluates on **nice diverse downstream datasets**. The average accuracy varies within a narrow band (64.1–65.4%), and multiple α values (0.05, 0.01) achieve similar performance. This suggests that PTA is robust to the choice of $\alpha$ in right range.
> >
> > Our evaluation does not just include COCO and Flickr30k. As shown in Table 1, we evaluate on a **diverse suite of seven downstream datasets** covering various domains and tasks, e.g., ImageNet (generic object recognition), Cars (fine-grained vehicle classification), and so on. During the evaluation, we do not provide any downstream labels or fine-tuning; we only use zero-shot evaluation to measure how well the strong model generalizes after fine-tuning on COCO with PTA. Across all these datasets, PTA consistently outperforms the baselines, demonstrating its generalization ability.
> >
> > > w4: Sensitivity to weak model quality: The method's success hinges on the weak model providing a meaningful similarity distribution. The ablation in A.5.2 shows robustness between B/32 and B/16, but what happens if the weak model is very weak or poorly calibrated? A systematic analysis of how the performance gap and quality of the weak model affect the strong model's final performance would provide valuable insights into the limits of this approach.
> >
> > **q-w4:** We agree that, in principle, PTA could fail if the weak model is too noisy or poorly calibrated, and we have therefore added a stress-test of weak model quality in App. A.5.3 of the revised paper. Instead of only comparing CLIP-B/32 vs. B/16, we fix the weak model to CLIP-B/32 and progressively degrade it by **injecting dropout** at inference time (rates 0.0, 0.1, 0.3, 0.5, 0.8), while keeping the strong model and all training hyperparameters fixed. As the dropout increases, the weak model’s average zero-shot accuracy over seven datasets drops from 61.0% (no dropout) to 3.54% (dropout 0.8, effectively near-random). In contrast, the strong model fine-tuned with PTA degrades only mildly: its average accuracy goes from 65.60% at dropout 0.0 to 64.63% at 0.3 (≤1 point variation for moderate degradation) and to 59.79% even at dropout 0.8, i.e., a relative drop of ≈9% despite a ≈94% relative drop in the weak model’s own performance. This systematic analysis shows that PTA is largely insensitive to moderate miscalibration or weakness of the teacher and degrades gracefully even when the weak model becomes extremely poor, indicating that the approach does not rely on a finely tuned or highly accurate weak model but only requires a teacher that is better than random. We summarize the results below and provide further details (per-dataset curves and discussion) in App. A.5.3 of the revised paper:
> >
> > | Dropout (weak) | Weak model avg acc (%) | Strong model avg acc (%) | Weak rel drop vs 0.0 | Strong rel drop vs 0.0 |
> > |---------------:|------------------------:|--------------------------:|----------------------:|------------------------:|
> > | 0.0           | 61.00                   | 65.60                     | 0.0%                  | 0.0%                    |
> > | 0.1           | 56.72                   | 65.57                     | 7.0%                  | 0.0%                    |
> > | 0.3           | 39.00                   | 64.63                     | 36.1%                 | 1.5%                    |
> > | 0.5           | 13.10                   | 61.23                     | 78.5%                 | 6.7%                    |
> > | 0.8           | 3.54                    | 59.79                     | 94.2%                 | 8.9%                    |

---

> > > ### Author Response · Authors · 2025-11-22
> > >
> > > Conversely, as long as the weak model is slightly better than random at ranking related vs. unrelated pairs, PTA can still suppress a fraction of hard false negatives and improve alignment, which is exactly what we observe in this ablation.
> > >
> > > > w5: Computational overhead: While the authors note a ~1.3x slowdown (Figure 6), this is a non-trivial cost, especially when considering large-scale pre-training. The requirement to compute the weak model's similarity for all in-batch negative pairs for every training step (to compute L\_PTA\_batch) adds a persistent overhead. A more detailed discussion on the scalability of this approach and potential optimizations (e.g., caching, using a much smaller proxy model as the "weak" model) would be beneficial.
> > >
> > > **q-w5:** We thank the reviewer for raising this important point about efficiency and scalability. We highly recommend to review what I have answered in a similar question for **w5** of reviewer **R9wB**. At the same time, we will expand the discussion in the main text.
> > >
> > > > w6: Clarity on loss formulation: The loss functions in Equations (12) and (13) could be clarified. The denominator sums over all B items in the batch, but the numerator only sums over those where l\_km^PTA = 1. This seems to encourage the model to assign high probability to any of the soft positives, which is a form of multiple-positive learning. This is a valid approach, but the connection to and difference from other multiple-positive losses could be discussed more explicitly.
> > >
> > > **q-w6:** We agree that the connection to multi-positive contrastive learning could be made more explicit. Our PTA loss indeed generalizes the standard InfoNCE loss to accommodate multiple positives per anchor, similar to supervised contrastive losses used in multi-label or multi-positive settings. The key difference is that PTA defines the positive set dynamically based on the permutation-calibrated weak similarities, rather than relying on fixed labels or heuristics.
> > >
> > > We will clarify this in the main text and add a discussion connecting our formulation to multi-positive losses.
> > >
> > > [1]. Mitigating Noisy Correspondence by Geometrical Structure Consistency Learning, CVPR 2024.
> > > [2]. Discovering Clone Negatives via Adaptive Contrastive Learning for Image-text Matching, ICLR 2025.
> > > [3]. Unlearning the Noisy Correspondence Makes CLIP More Robust, ICCV 2025.

---

### Official Review · Reviewer_6nzC · 2025-10-26

**Soundness:** 3
**Presentation:** 3
**Contribution:** 3
**Rating:** 6
**Confidence:** 4

**Summary:**

This paper proposes a weak-to-strong supervision framework to address the issue of false negatives in CLIP pretraining, which identifies and filters unreliable negatives based on permutation-based thresholding. Experiments on downstream tasks demonstrate the effectiveness of proposed method.

**Strengths:**

1. This paper proposes a weak-to-strong supervision framework to tackle the problem of false negatives in CLIP, which can filter unreliable negatives by estimating a null distribution of similarities through permutation-based thresholding and use weak models to provide training signals to guide strong models.
2. Experiments on downstream tasks including zero-shot classification demonstrate that proposed method can consistently improve the alignment ability of strong models.

**Weaknesses:**

1. While the paper dedicates significant space to its motivation in Sec. 3.1 (the false-negative problem in CLIP), this issue has been previously identified and explored by several existing works, such as PyramidCLIP [1], SoftCLIP [2], and CLIP-PSD [3]. The authors should properly include and discuss these relevant studies. Additionally, the "Related Work" section, currently placed in the appendix, should be integrated into the main body of the paper to ensure the completeness.
2. The fine-tuning dataset, COCO, has a limited data size. To further validate the scalability of the proposed method, it would be beneficial to include larger-scale public datasets such as CC3M [4], CC12M [5], YFCC [6], or LAION [7].
3. To provide a more comprehensive evaluation, it is recommended to include quantitative comparisons on the linear probing task. Furthermore, the addition of qualitative results—such as t-SNE visualizations, Grad-CAM heatmaps, or retrieval examples—would offer valuable intuitive insights into the model's behavior and strengths.
4. Regarding Eq. (15), what about the results when $\lambda >1$ ?

[1] Y. Gao, J. Liu, Z. Xu, J. Zhang, K. Li, and C. Shen. Pyramidclip: Hierarchical feature alignment for vision-language model pretraining. In NeurIPS, 2022.

[2] Y. Gao, J. Liu, Z. Xu, T. Wu, E. Zhang, W. Liu, J. Yang, K. Li, and X. Sun. SoftCLIP: Softer Cross-modal Alignment Makes CLIP Stronger. In AAAI, 2024.

[3] A. Andonian, S. Chen, and R. Hamid. Robust cross-modal representation learning with progressive selfdistillation. In CVPR, 2022.

[4] P. Sharma, N. Ding, S. Goodman, and R. Soricut. Conceptual captions: A cleaned, hypernymed, image alttext dataset for automatic image captioning. In ACL, 2018.

[5] S. Changpinyo, P. Sharma, N. Ding, and R. Soricut. Conceptual 12m: Pushing web-scale image-text
pre-training to recognize long-tail visual concepts. In CVPR, 2021.

[6] B. Thomee, D. A. Shamma, G. Friedland, B. Elizalde, K. Ni, D. Poland, D. Borth, and L.-J. Li. Yfcc100m:
The new data in multimedia research. Communications of the ACM, 2016.

[7] C. Schuhmann, R. Vencu, R. Beaumont, R. Kaczmarczyk, C. Mullis, A. Katta, T. Coombes, J. Jitsev, and
A. Komatsuzaki. Laion-400m: Open dataset of clip-filtered 400 million image-text pairs. In NeurIPS Workshop, 2021.

**Questions:**

See Weaknesses.

---

> ### Author Response · Authors · 2025-11-22
>
> We thank the reviewer for thoughtful feedback, constructive suggestions and supporting our work. Below we address each of your points in detail.
>
> **Weaknesses:**
>
> > w1: While the paper dedicates significant space to its motivation in Sec. 3.1 (the false-negative problem in CLIP), this issue has been previously identified and explored by several existing works, such as PyramidCLIP [1], SoftCLIP [2], and CLIP-PSD [3]. The authors should properly include and discuss these relevant studies. Additionally, the "Related Work" section, currently placed in the appendix, should be integrated into the main body of the paper to ensure the completeness.
>
> **q-w1**: We thank the reviewer for pointing out these relevant works. We will incorporate a discussion of PyramidCLIP, SoftCLIP, and CLIP-PSD into the main text to better reflect our contributions within the existing literature. We will also move the "Related Work" section from the appendix to the main body of the paper to enhance completeness and accessibility.
>
> > w2: The fine-tuning dataset, COCO, has a limited data size. To further validate the scalability of the proposed method, it would be beneficial to include larger-scale public datasets such as CC3M [4], CC12M [5], YFCC [6], or LAION [7].
>
> **q-w2**: We conducted an ablation experiment to see how well our method on a large-scale dataset (CC3M). Under same experimental settings, we compared standard supervised fine-tuning against PTA with the same FT4 setting (train from scratch). The results are as follows:
>
> | Method         | Data size | Avg. Acc. | ImageNet  | Cars     | DTD       | EuroSAT   | CIFAR-100 | CIFAR-10  | Food-101  |
> | -------------- | --------- | --------- | --------- | -------- | --------- | --------- | --------- | --------- | --------- |
> | SFT        | 3.0M      | 35.96     | 31.93     | 2.25     | 34.26     | 28.46     | 45.92     | 78.20     | 30.70     |
> | **PTA** | **3.0M**  | **37.21** | **34.27** | **2.59** | **34.89** | **28.75** | **47.63** | **78.55** | **33.79** |
>
> | run_name | coco_i2t_r@1 | coco_t2i_r@1 | coco_i2t_r@5 | coco_t2i_r@5 | flickr30k_i2t_r@1 | flickr30k_t2i_r@1 | flickr30k_i2t_r@5 | flickr30k_t2i_r@5 |
> |----------|----------------------|-----------------------|-----------------------|-----------------------|--------------------|--------------------|--------------------|--------------------|
> | SFT | 9.33 | 8.37 | 21.81 | 20.65 | 57.40 | 57.40 | 83.00 | 83.60 |
> | PTA  | 10.02 | 8.72 | 23.14 | 21.23 | 59.10 | 60.30 | 83.70 | 84.10 |
>
> PTA consistently outperforms the SFT across all datasets. In such a large-scale setting and relatively low-quality data, we witness PTA in fact demonstrates even larger improvements over our COCO results.
>
> > w3: To provide a more comprehensive evaluation, it is recommended to include quantitative comparisons on the linear probing task.
>
> **q-w3**: We thank reviewer for this suggestion to enhance our evaluation.
>
> We conducted linear-probing experiment on both COCO and Flickr30k, the permutation-calibrated PTA variant consistently outperforms the SFT across nearly every Recall@K axis, delivering stronger recall for both image-to-text and text-to-image queries. This demonstrates that PTA’s calibrated pseudo-labeling also provides gains in linear probe benchmarks.
>
> | Run | COCO R@1 (i→t / t→i) | COCO R@5 (i→t / t→i) | Flickr R@1 (i→t / t→i) | Flickr R@5 (i→t / t→i) |
> | --- | --- | --- | --- | --- |
> | PTA  | 22.19 / 20.99 | 41.62 / 40.30 | 78.90 / 78.40 | 94.80 / 94.70 |
> | SFT  | 21.71 / 19.69 | 40.84 / 38.67 | 78.60 / 76.60 | 95.20 / 94.50 |
>
> > w4: Furthermore, the addition of qualitative results—such as t-SNE visualizations, Grad-CAM heatmaps, or retrieval examples—would offer valuable intuitive insights into the model's behavior and strengths.
>
> **q-w4**: Thanks for the suggestion. We have added examples in Appendix. A.6 Visualization of our revised paper.
>
> > w5: Regarding Eq. (15), what about the results when $\lambda > 1$ ?
>
> **q-w5**: We thank the reviewer for pointing this out. We extended our ablation study to include $\lambda > 1$ values (specifically, $\lambda = 2.0, 5.0$) and observed that increasing $\lambda$ beyond 1.0 continues to yield performance improvements, with diminishing returns at higher values. The results are summarized in the table below.
>
> | Lambda (loss_alpha) | Average Accuracy |
> |---------------------|------------------|
> | 2.0                 | 65.66%           |
> | 5.0                 | 65.22%           |

---

### Official Review · Reviewer_vHJj · 2025-10-27

**Soundness:** 2
**Presentation:** 2
**Contribution:** 2
**Rating:** 4
**Confidence:** 4

**Summary:**

This paper proposes a weak-to-strong supervision framework called Permute-then-Adapt (PTA) to improve image–text alignment in contrastive models such as CLIP. The key motivation is that standard contrastive learning treats all unpaired samples as negatives, introducing false negatives when semantically related pairs are penalized. PTA uses a weak model to estimate semantic relatedness and employs a permutation-based thresholding mechanism to statistically determine which unpaired pairs are likely semantically relevant. These pairs are then used as soft positives to fine-tune a strong model. The method enriches binary supervision with soft, statistically grounded signals. Experiments on multiple benchmarks (COCO, Flickr30K, CIFAR, ImageNet, EuroSAT, etc.) show moderate improvements in both zero-shot classification and cross-modal retrieval tasks, with controlled robustness to batch size, α-level, and λ-weighting.

**Strengths:**

- The introduction of permutation-based thresholding provides a statistically grounded way to separate noise from meaningful weak signals, improving over heuristic thresholding.
- Experiments are broad, covering several datasets and extensive ablations on α, λ, and fine-tuning depth.

**Weaknesses:**

- The methodological novelty is relatively limited. The framework essentially combines soft-label supervision and weak-to-strong transfer with a permutation-derived threshold; the conceptual and empirical increment over prior soft alignment or pseudo-labeling methods is minor.
- The issue of false negatives in contrastive learning has already been widely recognized and addressed in many recent works; therefore, the empirical analysis in Section 3.1 mainly reiterates an established observation rather than offering new insights. More critically, the paper does not sufficiently articulate how its proposed PTA framework fundamentally differs from, or improves upon, prior baselines that also tackle the same problem through soft-labeling, pseudo-labeling, or adaptive thresholding.
- Gains may largely stem from controlled partial fine-tuning (FT4) rather than the permutation mechanism itself.

**Questions:**

- How does the proposed permutation-based threshold compare to simpler percentile-based or temperature-calibrated thresholds under identical fine-tuning settings? Is the improvement statistically significant?
- How sensitive is PTA to the domain and quality of the weak model? Would it still work if the weak model is trained on a different distribution (e.g., non-COCO data)?

---

> ### Author Response · Authors · 2025-11-22
>
> We thank the reviewer for thoughtful feedback and constructive suggestions. Below we address each of your points in detail.
>
> **Answers to weaknesses**
>
> > w1: The methodological novelty is relatively limited. The framework essentially combines soft-label supervision and weak-to-strong transfer with a permutation-derived threshold; the conceptual and empirical increment over prior soft alignment or pseudo-labeling methods is minor.
>
> **a-w1**: We would like to respectfully point out that our conceptual contribution goes beyond a minor increment over prior soft-alignment and pseudo-labeling work. Our setting is *post-hoc weak-to-strong refinement* of an pre-trained CLIP-style model, using only a weak model and no downstream labels, which is different from existing works that apply soft labels or pseudo-labels during pre-training or single-dataset training. Conceptually, PTA introduces a permutation-based null distribution over weak similarities and chooses the threshold via a user-controlled significance level $\alpha$, so $\alpha$ directly encodes an acceptable false-alarm rate rather than being a hand-tuned similarity cutoff. This yields a dataset-adaptive, statistically interpretable way to decide which non-ground-truth pairs are treated as additional positives, and we then integrate them through the standard CLIP loss, turning hard one-positive supervision into a multi-positive alignment objective without discarding the original labels.
>
> Empirically, PTA is not just another instance of “soft labels + weak-to-strong”: we directly compare against soft alignment, VL pseudo-labels, fixed thresholds, momentum distillation, and weak-to-strong augmentation under identical data, and architectures, and PTA remains the best across seven classification and two retrieval benchmarks. These improvements are consistent rather than one-off, and our ablations show that simpler percentile or fixed-Q variants underperform the permutation-calibrated PTA threshold.
>
> Finally, we explicitly highlight Reviewer zZyZ’s opinion that the permutation-based treatment of false positives is a **novel** component.
>
> > w-2: The issue of false negatives in contrastive learning has already been widely recognized and addressed in many recent works; therefore, the empirical analysis in Section 3.1 mainly reiterates an established observation rather than offering new insights. More critically, the paper does not sufficiently articulate how its proposed PTA framework fundamentally differs from, or improves upon, prior baselines that also tackle the same problem through soft-labeling, pseudo-labeling, or adaptive thresholding.
>
> **a-w2**: We agree that the existence of false negatives in contrastive learning is well-established, and we want to clarify that Section 3.1 is not intended as a claim of rediscovering this phenomenon. Its role is to quantify the problem in the specific problem we study: multi-modal learning and post-hoc weak-to-strong refinement on image-pair datasets without downstream labels. In this setting, we show (i) how the per-image false-negative rate grows sharply with batch sizes in CLIP, (ii) that these false negatives are concentrated among top-ranked “negatives” that are visually and textually close to the ground-truth pair, and (iii) that using a single global threshold leads to systematically different “acceptable” hardest-negative similarities across datasets and batch sizes. These observations are new in this particular weak-to-strong setting and are used to derive the design of our permutation-based calibration, rather than to claim a generic insight that false negatives exist.
>
> Regarding the difference from prior baselines, the fundamental distinction lies in **statistical grounding versus heuristic calibration**. Methods like *Soft Align* or *VL PseudoLabel* typically rely on fixed thresholds (e.g., cosine similarity > 0.3). As shown in our analysis, optimal thresholds shift significantly based on batch size and dataset distribution. PTA differs by generating a "null distribution" via permutation. This allows us to define positives based on statistical significance (rejecting the null hypothesis of independence at level $\alpha$) rather than absolute similarity scores. This ensures that PTA adapts automatically to different weak teachers or datasets without manual threshold tuning, a robustness confirmed by our ablation studies in the paper.

---

> > ### Author Response · Authors · 2025-11-22
> >
> > > w-3: Gains may largely stem from controlled partial fine-tuning (FT4) rather than the permutation mechanism itself.
> >
> > **a-w3**: We conducted a ablation study on CC3M where the only variable changed was the supervision signal, keeping other settings identical. Specifically, we compared standard supervised fine-tuning against PTA under the same FT4 setting. The results are as follows:
> >
> > | Method         | Data size | Avg. Acc. | ImageNet  | Cars     | DTD       | EuroSAT   | CIFAR-100 | CIFAR-10  | Food-101  |
> > | -------------- | --------- | --------- | --------- | -------- | --------- | --------- | --------- | --------- | --------- |
> > | SFT (w\o PTA)      | 3.0M      | 35.96     | 31.93     | 2.25     | 34.26     | 28.46     | 45.92     | 78.20     | 30.70     |
> > | **PTA** | **3.0M**  | **37.21** | **34.27** | **2.59** | **34.89** | **28.75** | **47.63** | **78.55** | **33.79** |
> >
> > Since the number of trainable layers and optimizer settings are identical, the performance difference reflects changes in the supervision signal rather than additional adaptation capacity. This supports the interpretation that PTA improves alignment by introducing statistically filtered soft positives, not by relying on deeper tuning.
> >
> > **Questions:**
> >
> > > q1: How does the proposed permutation-based threshold compare to simpler percentile-based or temperature-calibrated thresholds under identical fine-tuning settings? Is the improvement statistically significant?
> >
> > **a-q1**: To directly address how the permutation-based threshold compares to simpler alternatives, we fine-tuned the **same** weak-to-strong configuration while changing *only* the rule that classifies weak-model similarities as unsafe negatives. The two calibration strategies are:
> >
> > * **Naive percentile.** Threshold the raw weak logits at a fixed score, obtained by taking the 99th percentile o scores on true COCO pairs and reusing that cutoff throughout fine-tuning.
> > * **PTA.** Our method: derive the cutoff from the permuted null distribution of weak scores, using a user-specified significance level.
> >
> > The table below reports downstream classification accuracy averaged over 7 image classification benchmarks. The permutation-derived threshold obtains the best mean accuracy, indicating that calibrating against the permuted null distribution provides a cleaner set of “unsafe” negatives.
> >
> > | Method           | IN        | Cars      | DTD       | EuroSAT   | CIFAR100  | CIFAR10   | Food101   | Avg.      |
> > | ---------------- | --------- | --------- | --------- | --------- | --------- | --------- | --------- | --------- |
> > | Naive percentile | 62.99     | 58.74     | 43.51     | 49.36     | 66.06     | **89.68** | 81.79     | 64.59     |
> > | Permutation PTA  | **64.14** | **60.40** | **44.57** | 50.51     | **66.45** | 89.38     | **83.77** | **65.60** |

---

> > > ### Author Response · Authors · 2025-11-22
> > >
> > > > q2: How sensitive is PTA to the domain and quality of the weak model?
> > >
> > > **a-q2**: Regarding how PTA performs when the weak model is even smaller or weaker than CLIP-B/32. To directly address this, we have added a stress-test of weak model quality in App. A.5.3 of the revised paper. Instead of only comparing CLIP-B/32 vs. B/16, we fix the weak model to CLIP-B/32 and progressively degrade it by **injecting dropout** at inference time (rates 0.0, 0.1, 0.3, 0.5, 0.8), while keeping the strong model and all training hyperparameters fixed. As the dropout increases, the weak model’s average zero-shot accuracy over seven datasets drops from 61.0% (no dropout) to 3.54% at dropout 0.8 (effectively near-random). In contrast, the strong model fine-tuned with PTA degrades only mildly: its average accuracy goes from 65.60% at dropout 0.0 to 64.63% at 0.3 (less than 1 point variation under moderate degradation) and remains 59.79% even at dropout 0.8, i.e., a 8.9% relative drop despite a 94.2% relative drop in the weak model’s performance:
> > >
> > > | Dropout (weak) | Weak model avg acc (%) | Strong model avg acc (%) | Weak rel drop vs 0.0 | Strong rel drop vs 0.0 |
> > > | -------------: | ---------------------: | -----------------------: | -------------------: | ---------------------: |
> > > |            0.0 |                  61.00 |                    65.60 |                 0.0% |                   0.0% |
> > > |            0.1 |                  56.72 |                    65.57 |                 7.0% |                   0.0% |
> > > |            0.3 |                  39.00 |                    64.63 |                36.1% |                   1.5% |
> > > |            0.5 |                  13.10 |                    61.23 |                78.5% |                   6.7% |
> > > |            0.8 |                   3.54 |                    59.79 |                94.2% |                   8.9% |
> > >
> > > This systematic analysis shows that PTA is insensitive to moderate degradation of the teacher and degrades gracefully even when the weak model becomes extremely poor. In other words, CLIP-B/32 is used as a concrete, practically motivated weak supervisor (the smallest CLIP variant), but our results indicate that PTA does not rely on it being particularly strong or finely calibrated: **as long as the weak model is slightly better than random at ranking related vs. unrelated pairs, PTA can still suppress a fraction of hard false negatives and improve alignment**.
> > >
> > > > q3: Would it still work if the weak model is trained on a different distribution (e.g., non-COCO data)?
> > >
> > > **a-q3**: PTA is explicitly designed to be robust to the weak model’s domain and quality, and our main setup already uses an off-domain weak model. In all experiments, the weak teachers are CLIP-like models pre-trained on generic web data, not on COCO, while PTA is applied when fine-tuning a stronger CLIP on COCO, and we still see consistent gains over baselines.

---

### Official Review · Reviewer_R9wB · 2025-10-31

**Soundness:** 2
**Presentation:** 3
**Contribution:** 2
**Rating:** 4
**Confidence:** 3

**Summary:**

This paper introduces Diffusion Dataset Condensation (D2C), a framework of dataset distillation for diffusion models. The key idea is to construct a small yet information-rich synthetic sub-dataset that enables high-quality diffusion model training with only a fraction of the original data. D2C contains two phases: Select phase and Attach phase. The experiments show that only using 0.8% data the diffusion model can be trained from scratch and achieve a good performance.

**Strengths:**

1.  This is the first paper to formally study dataset condensation for diffusion models, whereas prior works (e.g., SRe2L, MTT, CAFE) targeted discriminative tasks like classification.

  2.  D2C achieves up to 233× faster training using only 0.8% of ImageNet data, while maintaining competitive FID (e.g., 4.3 at 40k steps).

**Weaknesses:**

1.  The proposed benchmark offers a controlled setting for studying weak supervision, but its novelty is limited since it builds on existing datasets rather than introducing new data. The weak signals derived from CLIP-B/32 may not reflect real-world noisy supervision, reducing external validity. Moreover, the benchmark and PTA framework are co-designed, which introduces potential self-validation bias and limits generalizability.

  2. The assumption here is that weak-model similarity scores adequately represent real-world weak supervision. In practice, models like CLIP-B/32 are already well-trained and produce relatively clean signals, which differ from the noisy, inconsistent annotations typically found in large-scale web data. As a result, the simulated weak supervision in this benchmark may underestimate the complexity and noise of real-world multimodal datasets, limiting the ecological validity of the findings.

 3. The experiments focus mainly on CLIP-style models (ViT-B/16 and ViT-B/32). It remains unclear whether the same improvements hold for other architectures or multimodal models (e.g., BLIP, ALIGN, or Qwen-VL), limiting generality.

 4. The PTA framework requires an additional forward pass of the weak model for every batch, leading to extra computational cost and potentially reduced scalability for large datasets. Although statistically consistent, the reported performance gains (+1–2%) are relatively modest, raising questions about the practical significance of the improvement versus the added training cost (≈1.3–1.4× slowdown).

**Questions:**

1. How does your benchmark differ fundamentally from existing datasets like COCO or Flickr30k beyond the weak-supervision setup?

2. Since the PTA framework and benchmark were co-designed, how do you ensure that the benchmark does not inherently favor your method over alternative weak supervision approaches?

The rest was stated in the weakness part.

---

> ### Author Response · Authors · 2025-11-22
>
> We thank the reviewer for thoughtful feedback and constructive suggestions. Below we address each of your points in detail.
>
> **Answers to weaknesses**
>
> > w1: The proposed benchmark offers a controlled setting for studying weak supervision, but its novelty is limited since it builds on existing datasets rather than introducing new data.
>
> **a-w1**: The benchmark is intentionally built on top of existing, widely used datasets rather than introducing new data. The aim is **not** to contribute a new dataset, but to define a *evaluation setting for weak-to-strong generalization in image–text alignment*. We will revise the paper to avoid language suggesting a “new benchmark” and instead describe this as a standardised evaluation protocol for weak-to-strong multimodal learning built on established datasets. We thank the reviewer for pointing this out. We thank the reviewer for pointing this out.
>
> > w2: The weak signals derived from CLIP-B/32 may not reflect real-world noisy supervision, reducing external validity. Moreover, the benchmark and PTA framework are co-designed, which introduces potential self-validation bias and limits generalizability.
>
> > w3: The assumption here is that weak-model similarity scores adequately represent real-world weak supervision. In practice, models like CLIP-B/32 are already well-trained and produce relatively clean signals, which differ from the noisy, inconsistent annotations typically found in large-scale web data. As a result, the simulated weak supervision in this benchmark may underestimate the complexity and noise of real-world multimodal datasets, limiting the ecological validity of the findings.
>
> **a-w2,3**: In our setup, CLIP-B/32 is used as a fixed, off-the-shelf teacher because it is the smallest publicly available model in the CLIP family, and thus a natural choice for an inexpensive weak supervisor [1]. **We do not claim CLIP-B/32 fully captures web-scale noise.** Our goal is to emulate a practical scenario where a cheaper or smaller multimodal model supervises a more capable one, rather than to cover all possible sources of real-world noisy supervision. The benchmark is therefore defined by standard datasets plus a fixed weak teacher, and PTA is one particular way of consuming these weak signals.
>
> To address the concern about co-design and potential self-validation, we emphasis that baselines and PTA are trained on the same COCO data with the *same* teacher scores (if used) and are evaluated on the same downstream tasks. Within this fixed setting, we compare PTA against several alternative ways of using the identical weak signals, so any performance gap comes from how the weak supervision is exploited, not from a special coupling between teacher and method.
>
> Regarding how our method performs when the weak model is even smaller or weaker than CLIP-B/32, we have added a stress-test of weak model quality in App. A.5.3 of the revised paper. Instead of only comparing CLIP-B/32 vs. B/16, we fix the weak model to CLIP-B/32 and progressively degrade it by **injecting dropout** at inference time (rates e.g., 0.0, 0.1, 0.3, 0.5, 0.8), while keeping the strong model and all training hyperparameters fixed. As the dropout increases, the weak model’s average zero-shot accuracy over seven datasets drops from 61.0% (no dropout) to 3.54% at dropout 0.8 (effectively near-random). In contrast, the strong model fine-tuned with PTA degrades only mildly: its average accuracy goes from 65.60% at dropout 0.0 to 64.63% at 0.3 (less than 1 point variation under moderate degradation) and remains 59.79% even at dropout 0.8, i.e., a ~9% relative drop despite a ~94% relative drop in the weak model’s performance:
>
> | Dropout (weak) | Weak model avg acc (%) | Strong model avg acc (%) | Weak rel drop vs 0.0 | Strong rel drop vs 0.0 |
> | -------------: | ---------------------: | -----------------------: | -------------------: | ---------------------: |
> |            0.0 |                  61.00 |                    65.60 |                 0.0% |                   0.0% |
> |            0.1 |                  56.72 |                    65.57 |                 7.0% |                   0.0% |
> |            0.3 |                  39.00 |                    64.63 |                36.1% |                   1.5% |
> |            0.5 |                  13.10 |                    61.23 |                78.5% |                   6.7% |
> |            0.8 |                   3.54 |                    59.79 |                94.2% |                   8.9% |

---

> > ### Author Response · Authors · 2025-11-22
> >
> > This systematic analysis shows that PTA is insensitive to moderate degradation of the teacher and degrades gracefully even when the weak model becomes extremely poor. In other words, CLIP-B/32 is used as a concrete, practically motivated weak supervisor but our results indicate that PTA does not rely on it being particularly strong or finely calibrated: as long as the weak model is slightly better than random at ranking related vs. unrelated pairs, PTA can still suppress a fraction of hard false negatives and improve alignment.
> >
> > > w4: The experiments focus mainly on CLIP-style models (ViT-B/16 and ViT-B/32). It remains unclear whether the same improvements hold for other architectures or multimodal models (e.g., BLIP, ALIGN, or Qwen-VL), limiting generality.
> >
> > **a-w4**: This work is specifically scoped to the **task** of image–text alignment, under a **contrastive learning objective** (image–text contrastive loss) and **CLIP-style duel-encoder architectures**, all evaluated on standard image–text alignment **benchmarks**, e.g., zero-shot classification and retrieval. Extending PTA to generative multimodal models such as BLIP or Qwen-VL would require substantial changes on all three dimensions: the task formulation (e.g., generative captioning or instruction following instead of pure alignment), the learning objective (next-token prediction losses rather than contrastive), and the datasets and evaluation protocols (e.g., caption quality or instruction adherence instead of retrieval/classification). Such an extension would therefore entail a non-trivial amount of additional work and is better treated as future work.
> >
> > > w5: The PTA framework requires an additional forward pass of the weak model for every batch, leading to extra computational cost and potentially reduced scalability for large datasets. Although statistically consistent, the reported performance gains (+1–2%) are relatively modest, raising questions about the practical significance of the improvement versus the added training cost (≈1.3–1.4× slowdown).
> >
> > **a-w5**: We thank the reviewer for this insightful comment. We view PTA as a *data-efficient* way to improve image-text alignment: rather than collecting more multimodal data, specifically we reuse the same batches and extract more supervision from them. To make this concrete, we ran a data-scaling study on CC3M and relate it to our COCO fine-tuning results. The student is CLIP-B/16 (last four ViT blocks trained from scratch, one epoch); the weak model is CLIP-B/32. All baseline are trained under the identical settings, varying only the number of CC3M pairs used (1.0 M, 2.0 M, or 3.0 M). The results are as follows:
> >
> > | Method | Data size (pairs) | Avg. acc. (%) | ImageNet | Stanford Cars | DTD | EuroSAT | CIFAR-100 | CIFAR-10 | Food-101 |
> > | --- | --- | --- | --- | --- | --- | --- | --- | --- | --- |
> > | SFT-1 | 1.0 M | 31.65 | 24.09 | 1.34 | 30.16 | 28.93 | 41.60 | 75.45 | 19.96 |
> > | SFT-2 | 2.0 M | 34.56 | 29.90 | 1.79 | 32.34 | 28.27 | 44.29 | 78.90 | 26.43 |
> > | SFT-3 | 3.0 M | 35.96 | 31.93 | 2.25 | 34.26 | 28.46 | 45.92 | 78.20 | 30.70 |
> > | **PTA (ours)** | **3.0 M** | **37.21** | **34.27** | **2.59** | **34.89** | **28.75** | **47.63** | **78.55** | **33.79** |
> >
> > Two observations are as follows:
> >
> > 1. **Diminishing returns from more data.**
> >    Scaling SFT from 1.0 M to 2.0 M examples adds 2.91 points, while the next 1.0 M (2.0 M to 3.0 M) yields 1.40 points. On average, each additional million CC3M pairs delivers roughly 2.15 absolute points on the seven-dataset average. This confirms that in this setting the task is already quite **challenging**: even adding another million image–text pairs only moves the needle by a few points.
> >
> > 2. **PTA achieves a comparable gain *without new data*.**
> >    On the same 3 M examples, PTA delivers 1.25 points over SFT-3. Using the empirical slope above, this gain is equivalent to what SFT would obtain by adding roughly 0.6 M *extra* CC3M pairs, yet PTA simply reuses the existing minibatches and extracts more structure from their weak similarities. In other words, PTA behaves like a data-efficient regulariser: it gives the kind of improvement you would normally need a much larger dataset for, but it works on the same fixed training set.
> >
> > This perspective also clarifies why our “+1–2%” improvements on COCO are not marginal. COCO provides <0.1 M clean captioned images, and our fine-tuning is done in a strict generalisation setting (no downstream labels, only zero-shot evaluation). In such settings, an extra 1–2 points is precisely the kind of gain that required *millions* of additional CC3M pairs in the large-scale experiment above. PTA is thus not just a small numerical bump; it is a data-efficient way to reduce false negatives and improve alignment when collecting more high-quality multimodal data is expensive or simply not an option.

---

> > > ### Author Response · Authors · 2025-11-22
> > >
> > > **Answers to specific questions**
> > >
> > > > q1: How does your benchmark differ fundamentally from existing datasets like COCO or Flickr30k beyond the weak-supervision setup?
> > >
> > > **a-q1**: We do not introduce a fundamentally new dataset. Instead, our contribution is to *re-purpose* existing datasets into a weak-to-strong evaluation **protocol**. Concretely, we (i) fix a weak teacher model that provides **weak** supervision, (ii) define a permutation-based procedure to turn these scores into calibrated labels with error control, and (iii) evaluate the resulting strong model on a shared suite of downstream datasets without using any downstream labels for training. All methods in our study, including PTA and the baselines, use exactly the identical fine-tuning settings.
> > >
> > > > q2: Since the PTA framework and benchmark were co-designed, how do you ensure that the benchmark does not inherently favor your method over alternative weak supervision approaches?
> > >
> > > **a-q2**: In our setup, PTA and all baselines in our experiments share the following:
> > > * uses the **same weak teacher scores** from CLIP-B/32 if used,
> > > * is trained on the **same COCO data** with the same strong backbone, optimizer, and augmentations, and
> > > * is evaluated on the **same downstream suite** with identical zero-shot settings.
> > > For examples, fixed thresholding, soft-align, and PTA all use the same weak teacher scores from CLIP-B/32 and the same COCO training data. The only difference is *how* each method exploits these weak signals during training. Therefore, any performance gap between PTA and the baselines must come from the method of consuming the weak supervision, not from a special coupling between teacher, data, and method.
> > >
> > > [1] Burns, C., Izmailov, P., Kirchner, J. H., Baker, B., Gao, L., Aschenbrenner, L., ... & Wu, J. (2023). Weak-to-strong generalization: Eliciting strong capabilities with weak supervision.

---

### Meta-Review · Area_Chair_ynbf · 2025-12-15

**Summary:**

The paper proposes Permute-Then-Adapt (PTA), a framework leveraging permutation-based thresholding to address false negatives in contrastive learning. While one reviewer appreciated the statistical grounding of the method, the overall evaluation leans negative with scores of 4, 4, 4, and 6. Reviewers consistently raised concerns regarding the limited methodological novelty compared to existing soft-label approaches, the practical drawbacks of the computational overhead (~1.3-1.4x slowdown), and the initial lack of comparisons with noisy-correspondence baselines. Although the authors provided a comprehensive rebuttal addressing robustness and data efficiency, the consensus remains that the performance gains are incremental relative to the added complexity, leading to a decision to reject.

**Reviewer Concerns:**

- Addressed: The rebuttal effectively addressed concerns regarding the method's sensitivity to weak model quality, provided the requested comparisons with baselines like AdaCL, and offered a strong justification for the computational cost from a data-efficiency perspective.
- Outstanding: The primary outstanding concerns are the perceived limited conceptual novelty and the practical scalability of the method, as the computational overhead for every batch remains a significant barrier for adoption in large-scale pre-training.

**Reviewer Scores:**

- Reviewer R9wB (4 -> 4): The concerns about computational overhead and real-world applicability likely persist despite the clarifications.
- Reviewer vHJj (4 -> 4): The criticism regarding "limited novelty" is fundamental and typically hard to overturn via rebuttal.
- Reviewer 6nzC (6 -> 6): The rebuttal supports the reviewer's original positive stance on the experiments.
- Reviewer zZyZ (4 -> 5): The authors have addressed this reviewer's specific requests for stress tests and missing baselines, which would likely have warranted a score increase.

---

### Decision · Program_Chairs · 2026-01-26

Reject